January 16, 2018

# Generalized global symmetries and holography

Diego M. Hofman[1,*] and Nabil Iqbal[2,†]

*¹Institute for Theoretical Physics, University of Amsterdam,
Science Park 904, Postbus 94485, 1090 GL Amsterdam, The Netherlands*
*²Centre for Particle Theory, Department of Mathematical Sciences,
Durham University, South Road, Durham DH1 3LE, UK*

## Abstract

We study the holographic duals of four-dimensional field theories with 1-form global symmetries, both discrete and continuous. Such higher-form global symmetries are associated with antisymmetric tensor gauge fields in the bulk. Various different realizations are possible: we demonstrate that a Maxwell action for the bulk antisymmetric gauge field results in a non-conformal field theory with a marginally running double-trace coupling. We explore its hydrodynamic behavior at finite temperature and make contact with recent symmetry-based formulations of magnetohydrodynamics. We also argue that discrete global symmetries on the boundary are dual to discrete gauge theories in the bulk. Such gauge theories have a bulk Chern-Simons description: we clarify the conventional 0-form case and work out the 1-form case. Depending on boundary conditions, such discrete symmetries may be embedded in continuous higher-form symmetries that are spontaneously broken. We study the resulting boundary Goldstone mode, which in the 1-form case may be thought of as a boundary photon. Our results clarify how the global form of the field theory gauge group is encoded in holography. Finally, we study the interplay of Maxwell and Chern-Simons terms put together. We work out the operator content and demonstrate the existence of new backreacted anisotropic scaling solutions that carry higher-form charge.

---

*Electronic address: d.m.hofman@uva.nl

†Electronic address: nabil.iqbal@durham.ac.uk

## I. INTRODUCTION

In this paper, we will discuss the manifestation of *generalized global symmetries* in $d$-dimensional quantum field theories with holographic duals. In the language of [1], a continuous generalized global $p$-form symmetry is associated with the conservation of an antisymmetric tensor current of rank $p + 1$. In form language, this conservation law can be written with the help of the Hodge $\star$ operator as the existence a co-closed $p + 1$ form $J$

$$d \star J = 0 \,. \tag{1.1}$$

The existence of such $p + 1$-forms guarantees the existence of conserved quantities integrated over $d - p - 1$ surfaces as

$$Q = \int_{\mathcal{M}_{d-p-1}} \star J \,. \tag{1.2}$$

In this language, 0-form symmetries give rise to standard conserved currents yielding charges as we integrate them over all of space at fixed time.

We can think about these charges is the following way: they count the number of charged objects piercing the surface $\mathcal{M}_{d-p-1}$. These objects are always $p + 1$ dimensional. This is consistent with the fact that, provided a background $p + 1$-gauge field $A$ for the current $J$, charged objects couple to it by introducing a term in the action of the form:

$$\delta S = iq \int_{\mathcal{C}_{p+1}} A \,, \tag{1.3}$$

where $\mathcal{C}_{p+1}$ is the world-volume of the charged objects. For familiar 0-form symmetries these objects have a 1-dimensional world-volume and we conclude that particles are the natural charged objects of 0-form symmetries. This fact allows for the large number of constructions of local theories enjoying 0-form symmetries: because particles are the quanta of local fields, there is always a quantum field theory description available for these systems.

The situation is manifestly different for higher form symmetries. Consider a 1-form symmetry. The natural objects charged under the symmetry are strings. An analog description in this case to the quantum field theory would be constituted by a theory where the fundamental operators live in loop space [2]. A more concrete way of saying this is that operators that create strings on 1 dimensional contour $d\mathcal{C}$ are non-local operators $\Phi[d\mathcal{C}]$, where $dC$ is the boundary of the 2 dimensional string world-volume $\mathcal{C}$. Unfortunately these objects are notoriously hard to work with and a systematic construction of these theories is not known[1].

---

[1] See, however, [3] for a recent discussion.

There exist an alternative description, however, of theories with 1-form symmetries. The main idea is to consider local quantum field theories where the extended objects are composite non local operators and not the fundamental degrees of freedom. Because we were hoping to construct a theory of extended objects, somehow the local degrees of freedom described by local operators in the quantum field theory should not be entirely physical: instead only their assembly in terms of the physical non-local objects should be. This is nothing but the standard construction of gauge theories. In this setup 't Hooft lines and Wilson lines are the physical gauge invariant operators constructed from the gauge fields. We review this gauge theory construction in terms of their physical 1-form symmetries in the next section.

The advantage of recasting the theory of extended objects in quantum field theory language comes, however, at a price: the introduction of unphysical symmetries in the form of a gauge group and gauge constraints in the Hilbert space of states. This fact obscures substantially the role of the physical symmetries that we were interested in in the first place. This is particularly cumbersome when the physical continuous symmetries are broken (spontaneously or explicitly) down to finite subgroups. An example of this difficulty concerns the discrete symmetries of $\mathcal{N} = 4$ Super Yang Mills (SYM), which we discuss in section III.

Luckily, we know of yet another description of gauge theories: the one given by the holographic principle [4] in terms of a dual gravitational $AdS$ bulk. This description is manifestly gauge invariant and it allows for a more clear interpretation of physical symmetries. Surprisingly, to our knowledge, the problem of understanding the precise holographic description of continuous higher form symmetries and their discrete subgroups has not been attacked systematically. It is the main purpose of this paper to fill this gap in the literature.

The recent paper [5] also studies generalized symmetry in holography from the point of view of magnetohydrodynamic applications.

## A.  Two concrete examples

In this work, we will mostly consider various realizations of 2-form currents $J$. While we will focus mostly on holographic aspects of these theories, it is first useful to discuss some standard QFT constructions. A thorough discussion can be found in [1]. Formal aspects related to the phase structure and symmetry algebra of the continuous abelian case will be discussed in a forthcoming publication [6].

In this short section we consider two examples.

### 1. Free electromagnetism

Consider first free electromagnetism in four-dimensions, with no matter and a (precisely marginal) gauge coupling $g$. This is conventionally written in terms of a 1-form gauge field $A_e$ with associated field strength $F = dA_e$. This theory possesses two different 2-form currents:

$$J_e \equiv \frac{1}{g^2} F, \qquad J_m \equiv g^2 G \tag{1.4}$$

with

$$G = \frac{1}{g^2} \star_4 F. \tag{1.5}$$

$J_m$ counts magnetic flux lines; its conservation is equivalent to the Bianchi identity associated with the existence of the potential $A_e$, i.e. the non-existence of dynamical magnetic monopoles. $J_e$ counts electric flux lines; its conservation is equivalent to the source-free Maxwell equation, i.e. the non-existence of dynamical electric charges. Therefore this theory has not one but two independent 2-form currents associated to 1-form symmetries. The objects that are charged under these symmetries are Wilson lines (electric) and 't Hooft lines (magnetic).

An interesting feature of this theory is now manifest. Note that $J_e \sim dA_e$ as a consequence of the Bianchi identity: the nonlinear realization of the 1-form symmetry associated to the 2-form current $J_e$ means that it is spontaneously broken, and we may think of the electric photon $A_e$ as being its gapless Goldstone mode [1]. We could equivalently have formulated the theory in terms of a magnetic photon $A_m$, in which case we would have concluded that $J_m \sim dA_m$ as a consequence of the absence of electric charges; thus in the free photon phase of Maxwell electrodynamics both generalized global symmetries are spontaneously broken.

The presence of a second current and spontaneous symmetry breaking end up being intimately related. Interestingly a variant of this phenomenon is quite general and it occurs in all dimensions for all generalized global symmetries. Whenever a symmetry is spontaneously broken it can be nonlinearly realized as $J \sim d\theta$ for some goldstone $\theta$. But this immediately implies that the current $\star_d J$ is also conserved in this phase. This is just a trivial consequence of the fact that "monopole" configurations become gapped in the symmetry broken phase and at low energies they can't break the emergent conservation of $\star_d J$. The holographic manifestation of this point will appear in this paper, and a more complete discussion and the relation to critical points will be presented in [6].

One may now want to introduce sources for the symmetry currents, i.e. to gauge the 1-form global symmetry above with a background 2-form source. As both currents are related[2]

---

[2] A way of making this more manifest corresponds to writing the two currents as self-dual and anti-self-dual

by $\star_4$, a single background 2-form gauge field $b_e$ for the electric current is sufficient, and the source for the magnetic current is effectively constructed as:

$$b_m = \frac{1}{g^2} \star b_e \tag{1.6}$$

Now, in a symmetry broken phase the covariant derivative is not linear on the Goldstone fields $A_{e,m}$. Because of this, the currents must be improved[3] to preserve gauge invariance as:

$$J_e \to \frac{1}{g^2} (F - b_e) , \qquad J_m \to g^2 (G - b_m) . \tag{1.7}$$

The gauge transformations are given for the electric and magnetic symmetries as:

$$e : \quad A_e \to A_e + \varphi_e , \quad b_e \to b_e + d\varphi_e , \qquad m : \quad A_m \to A_m + \varphi_m , \quad b_m \to b_m + d\varphi_m . \tag{1.8}$$

It is immediate from the above expressions that electric gauge transformations introduce physical changes of the magnetic gauge potential and vice-versa. This will make the conservation equations anomalous as we observe below.

We may now write an action for our theory in the presence of background fields in the electric formulation as

$$S = -\frac{1}{2g^2} \int (F - b_e) \wedge \star_4 (F - b_e) \tag{1.9}$$

or equivalently[4] in the magnetic formulation as

$$S = -\frac{g^2}{2} \int (G - b_m) \wedge \star_4 (G - b_m) \tag{1.10}$$

The action above is universal from the low energy point of view: it is the effective action for the Goldstones given the symmetries are spontaneously broken. Therefore, regardless of the UV completion of the theory the Maxwell action is the universal description in the symmetry broken phase.

From this is obvious that the currents (1.7) may be obtained by taking functional derivatives with respect to $b_e$ and $b_m$ appropriately. Notice that independent of the choice of fundamental fields ($A_e$ or $A_m$) the action can be made to depend on either $b_e$ or $b_m$, but one cannot write an action that depends on both sources in a way that both symmetries in (1.8) are locally realized on the fundamental fields.

---

components. In this case the currents are independent but shortened by the self-duality constrains. In Lorentzian signature this requires complexifying the gauge field.

[3] Note this is simply the higher form generalization of the usual form of the superfluid current $j = v^2(d\theta - a)$ for a spontaneously broken 0-form symmetry, where $\theta$ is the Goldstone and $a$ the external source.

[4] These two actions are equal using (1.5) and (1.6). However if we perform electric-magnetic duality (1.9) in the usual manner, we obtain (1.10) up to a contact term $b_m^2$: this is another way to understand the difference in anomaly structure described below.

However, the equations of motion are different depending on the choice of fundamental fields. This is the origin of an anomaly that breaks one of the conservation laws in the presence of background fields. In particular, taking $A_e$ to be the fundamental field we obtain:

$$d \star_4 \frac{1}{g^2} (F - b_e) = 0 \,, \qquad d \star_4 g^2 (G - b_m) = H_e \,, \tag{1.11}$$

with

$$H_e = db_e \,. \tag{1.12}$$

In (1.11), the first equation is the equation of motion, while the second equation is just the Bianchi identity written in a useful way. One can see immediately that the electric 2-current remains conserved in this case while the magnetic 2-current conservation is broken by the curvature of the background gauge field $b_e$. Note also that from the first equation one can interpret $\frac{1}{g^2} d \star_4 b_e$ as (the Hodge dual of) a fixed external electric charge current that acts as a source for the gauge field.

Conversely, if one takes the magnetic degrees of freedom to be fundamental, it is the electric 2-current conservation that is broken:

$$d \star_4 g^2 (G - b_m) = 0 \,, \qquad d \star_4 \frac{1}{g^2} (F - b_e) = -H_m \,, \tag{1.13}$$

with

$$H_m = db_m \,. \tag{1.14}$$

In conclusion, in the presence of a source for the magnetic 2-current $J_m$, the electric 2-current $J_e$ is no longer quite conserved. As its non-conservation is given by a fixed external source (and not by a dynamical operator), this is an *anomaly*. More precisely, it is a mixed anomaly preventing the simultaneous gauging of the electric and magnetic 2-currents. Note that this entire structure has an analogue in a massless scalar in two dimensions, which has conventional momentum and winding 1-currents that have a similar mixed anomaly.

## 2. Electromagetism coupled to electrically charged matter

We now turn to a different theory: consider usual Maxwell electrodynamics written in terms of an electric gauge potential $A$, coupled to light electrically charged matter that we schematically represent by $\phi$

$$S = -\frac{1}{2g^2} \int dA \wedge \star_4 dA + S[\phi, A] \,. \tag{1.15}$$

The action $S[\phi, A]$ represents the matter action minimally coupled to $A$ in the usual (0-form) gauge invariant manner.

In this case the $A$ equations of motion are:

$$\frac{1}{g^2}d \star_4 F = \star_4 j[\phi, A], \qquad dF = 0,$$

(1.16)

with

$$F = dA \qquad j[\phi, A] \equiv \frac{\delta S[A, \phi]}{\delta A}.$$

(1.17)

The operator $j[\phi, A]$ denotes the usual electrical 1-current that we gauge in the $\phi$ theory to couple it to $A$.

While $J_m \sim \star_4 dA$ is still conserved (as there are still no magnetic monopoles), $J_e$ is not, as electric field lines can now end on electric charges. We saw above that spontaneous breaking of the magnetic current implied conservation of the electric current. Thus we can conclude that the magnetic symmetry is no longer spontaneously broken. Note also that the coupling to electrically charged matter means that the magnetic presentation of the action (1.10) and realization of the magnetic symmetry (1.8) is no longer simple.

Thus to access $J_m = \star_4 F$ we now couple a source minimally in a different way as:

$$S = \int -\left( \frac{1}{2g^2} dA \wedge \star_4 dA + b_m \wedge F \right) + S[\phi, A].$$

(1.18)

As required we recover $J_m$ by varying with respect to the source $b_m$. Notice we can integrate by parts the source term to rewrite

$$S = \int -\left( \frac{1}{2g^2} dA \wedge \star_4 dA + A \wedge H_m \right) + S[\phi, A].$$

(1.19)

We see clearly that the field strength for $b_m$ is electrically charged under the gauged 0-form symmetry and introduces a background, modifying the electric current as

$$j[\phi, A] \to j[\phi, A] = \frac{\delta S[A, \phi]}{\delta A} + \star_4 H_m.$$

(1.20)

Note that we no longer refer to this as an anomaly as the 1-form electric symmetry was already broken explicitly by the $\phi$ sector of the theory.

Thus, this theory now contains only a *single* 2-form conserved current independent of anomalies and it is so in a different class than the first example. We note also that within a perturbative treatment such theories are not conformal, as $g$ runs logarithmically. If the IR is weakly coupled and we can ignore electric charges, we will obtain an enhancement of symmetry to the electric sector in the infrared, reproducing the above discussion. If the theory becomes strongly coupled it could develop a gap (e.g. by developing a superconducting condensate) in which case the magnetic symmetry is preserved, with all charged excitations above the gap. If the theory remains gapless but strongly coupled, we will argue in [6] that the electric symmetry is once again emergent at the fixed point. This connection between

the structure of conserved currents and (non)-conformality is borne out in the holographic model and it is one of the main points discussed in this work.

We thus note that the single characteristic that allows one to identify the set of theories that one might call "U(1) gauge theories coupled to matter in four dimensions" is actually the existence of a single conserved 2-current representing conserved magnetic flux. No mention of gauge symmetry is needed in this description. In [7] (see also earlier work in [8, 9]) the hydrodynamic theory of such a system at finite temperature was developed and shown to be equivalent to a generalized form of relativistic magnetohydrodynamics. See also [10] for a recent discussion of magnetohydrodynamics in the conventional formulation.

## B.  Plan for this paper

Here we outline the contents of the following sections in this paper.

In section II we consider a 5 dimensional AdS bulk theory of Maxwell type for a 2-form gauge potential $B$. We show this theory possesses a single 2-form current dual to $B$. Furthermore we discuss the identification of sources and responses in this theory. It turns out that that the source presents a logarithmic ambiguity dual to the renormalization group running of a marginal operator in the dual theory. We also show that physical quantities in this theory can be defined in terms of a renormalization group invariant scale associated to a Landau pole. This agrees with the comments above: a theory with a single 2-form current presents a logarithmic running.

Then we consider this theory at finite temperature. We calculate the charge susceptibility as well as the diffusion constant from quasinormal modes. They are both seen to depend on the Landau pole scale. The resistivity associated to the transport of electric charges is also computed and found to satisfy an Einstein relation with the above quantities. We last consider numerically the emergence of the boundary photon degree of freedom at energies much higher than the temperature.

In section III the origin of Chern-Simons theories in the bulk of $AdS$ as a consequence of symmetry breaking of continuous symmetries down to discrete subgroups is discussed. We first review the situation for usual 0-form symmetries. We make a detailed distinction between spontaneous symmetry breaking and explicit symmetry breaking and explain that from the point of view of the bulk this phenomenon depends only on boundary conditions and not on the bulk action. We recover the statements made in the previous section: conformal fixed points present in the IR either two spontaneously broken symmetries or their disappearance from low energy physics.

This discussion is extended to the case of 1-form symmetries and connections with the holographic description of $\mathcal{N} = 4$ Super Yang Mills with U(N) or SU(N) gauge groups are

outlined. The role of discrete symmetries in this case is highlighted.

Lastly, in section IV, we combine the elements of previous sections. We discuss the operator content of the theory. In the case where a relevant operator is present in the theory, in addition to the conserved currents, we obtain new infrared geometries corresponding to fixed points that break Lorentz invariance, enjoy anisotropic scaling and are a generalization of Lifshitz geometries with a new dynamical scaling exponent $\xi$. These solutions are of physical relevance for $\mathcal{N} = 4$ Super Yang Mills and its $\frac{1}{N}$ corrections.

We end with conclusions where our results our discussed and future directions are suggested. Finally we add three appendices with our conventions, a discussion of hydrodynamic results for diffusive modes using the technology from [7] and a review of the spectrum of allowed line operators in $U(N)$ gauge theory.

## II.   MAXWELL TYPE HOLOGRAPHY

We now turn to holography. Consider an antisymmetric 2-form field $B$ propagating in a 5d bulk. We would like the bulk action to be invariant under a gauge redundancy parametrized by a 1-form $\Lambda$:

$$B \rightarrow B + d\Lambda \ . \tag{2.1}$$

The simplest action one can write for this theory is the Maxwell-type action[5]

$$S[B] = \frac{1}{6\gamma^2} \int d^5 x_{\mathcal{M}} \ \sqrt{-g} H_{MNP} H^{MNP} \tag{2.2}$$

where $H = dB$ is the field strength of the 2-form $B$. In this section we will study the physics of this system propagating on a fixed asymptotically AdS$_5$ background.

We note that in 5 bulk dimensions this action is the Poincare dual of a conventional 1-form gauge field $A$, related to B via $dB \sim \star_5 dA$. Of course the physics of a conventional Maxwell gauge field is very well-studied in AdS/CFT. It is well-understood that the gauge field is dual to a normal one-form current $j^\mu$. In the absence of bulk objects that are charged under $B$ (or $A$) the calculational difference between these two systems results entirely from a difference in boundary conditions at infinity. We will see that this will result in a very different boundary interpretation.

We choose coordinates so that $r$ is the holographic direction and the AdS boundary is at $r \rightarrow \infty$. One expects that the boundary value of the $B$ field is related to the field-theory source as

$$B_{\mu\nu}(r \rightarrow \infty) = b_{\mu\nu} \tag{2.3}$$

---

[5] In this section we use $M, N, P$ indices for the bulk, $\mu, \nu, \rho$ for the boundary and $i, j, k$ for the spatial components of the boundary.

As we will see, this equation is actually ambiguous: logarithmic divergences as we approach the boundary will require us to interpret it carefully. Nevertheless, by taking functional derivatives of the on-shell action in the standard manner, we find that the field-theory current is related to the boundary value of the field-strength of $H$:

$$\langle J^{\mu\nu}\rangle = -\frac{1}{\gamma^2} \lim_{r\to\infty} \sqrt{-g}H^{r\mu\nu} \tag{2.4}$$

The bulk equation of motion for $B$ is

$$\partial_M \left(\sqrt{-g}H^{MNP}\right) = 0, \tag{2.5}$$

## A. Vacuum correlations and marginal deformations

Consider first a field theory that is Lorentz-invariant but not necessarily conformal, with a conserved antisymmetric current $J^{\mu\nu}$. We begin by studying the theory on flat 4d Euclidean space $(\tau, x^i)$. Recall that $J$ is a dimension-2 operator. Conservation of the current and anti-symmetry together imply that the vacuum momentum-space correlator must take the form

$$\langle J^{\mu\nu}(k)J^{\rho\sigma}(-k)\rangle = \left(-\frac{1}{k^2}\left(k^\mu k^\rho g^{\nu\sigma} - k^\nu k^\rho g^{\mu\sigma} - k^\mu k^\sigma g^{\nu\rho} + k^\nu k^\sigma g^{\mu\rho}\right) + \left(g^{\mu\rho}g^{\nu\sigma} - g^{\mu\sigma}g^{\nu\rho}\right)\right) f\left(\frac{|k|}{\Lambda}\right) \tag{2.6}$$

where $f$ is a dimensionless function and $\Lambda$ is some scale. If we were studying a conformal field theory, then $f$ would be a constant.

With no loss of generality, we may rotate the momentum to point entirely in the $\tau$ direction and call it $\Omega$. The only nonzero components of the correlator are now in the tensor channel, i.e. the full information is captured by

$$\langle J^{ij}(\Omega)J^{ij}(-\Omega)\rangle = f\left(\frac{\Omega}{\Lambda}\right) \tag{2.7}$$

The information of the correlator is captured by the scalar function of momentum $f$.

Let us now turn to holography. We work on pure Euclidean AdS$_5$ with unit radius:

$$ds^2 = \frac{dr^2}{r^2} + r^2\left(d\tau^2 + dx^i dx^i\right) \tag{2.8}$$

We parametrize the bulk field as

$$B_{ij}(r,\tau) = \sigma_{ij}\beta(r)e^{i\Omega\tau} \tag{2.9}$$

with $\sigma_{ij}$ a constant polarization tensor. (2.5) then becomes simply:

$$\partial_r \left(r\partial_r\beta(r)\right) - \frac{\Omega^2}{r^3}\beta(r) = 0 \tag{2.10}$$

Let us first study the asymptotic behavior of this equation as $r \to \infty$. Expanding the solutions at infinity we find:

$$\beta(r \to \infty) \sim \hat{b} - \gamma^2 J \log r . \tag{2.11}$$

At the moment $\hat{b}$ and $J$ are just expansion coefficients, although $J$ is so named because from (2.4) we see that when multiplied by the polarization tensor it is equal to the current $J\sigma_{ij} = J_{ij}$. From (2.3) it appears that $\hat{b}$ should be interpreted as the source: however we see that actually its value is *ambiguous* and runs logarithmically as we take $r \to \infty$. Note that $J$ is unambiguous.

The running of $\hat{b}$ indicates that the physics depends on the value of $r$ at which the boundary condition is applied. Thus the dual theory is not actually conformal. This arises because $J$ is a dimension 2 operator, and the double-trace coupling $J^2$ in the boundary is *marginal* but not exactly so: depending on its sign the running is marginally relevant or irrelevant, and it is this logarithmic running that we are seeing here. Precisely the same phenomenon happens for pure Maxwell theory for a 1-form gauge field on AdS$_3$ (where it is double-trace of the normal one-form current $j^\mu$ that is marginal) and was discussed in detail in [11] (see also [12, 13] for earlier study in the context of a scalar field).

We briefly summarize the discussion of [11] here. Consider deforming the CFT by a double-trace coupling $\frac{1}{\kappa}J^2$. Via the usual holographic dictionary [12, 14], this modifies the relation between the asymptotic values of the bulk field and the source $b$ to read:

$$\beta(r_\Lambda) - \gamma^2 \frac{J}{\kappa} = b \tag{2.12}$$

where the boundary condition is now applied at a particular scale $r_\Lambda$, and where we have traded the (ambiguous) expansion coefficient $\hat{b}$ for the (well-defined but radially varying) value of the bulk field $\beta(r_\Lambda)$ itself. The boundary condition is thus labeled by two parameters $r_\Lambda$ and $\kappa$.

However, the boundary condition can equivalently be applied at a different scale $r'_\Lambda \equiv \lambda r_\Lambda$ provided we also take the double trace-coupling to transform as

$$\frac{1}{\kappa'} + \log \lambda = \frac{1}{\kappa} . \tag{2.13}$$

This is precisely the logarithmic running of a marginal coupling. This means that dimensional transmutation should occur, and all observables should depend not on $\kappa$ and $r_\Lambda$ separately, but rather only on the RG-invariant scale

$$r_\star \equiv r_\Lambda e^{\frac{1}{\kappa}} . \tag{2.14}$$

For $\kappa > 0$, this scale is in the UV. To understand this, it is helpful to consider ordinary QED coupled to dynamical matter, which is a theory with the same symmetries as the holographic

theory we are studying here and where $\kappa$ would be identified with $e^2$, the electromagnetic coupling. $r_\star$ is then the Landau pole at which the theory breaks down.

We now explicitly compute the correlator, which is defined to be the ratio of the source and response:

$$f(\Omega) \equiv \frac{J}{b} = \frac{1}{\frac{\beta(r_\Lambda)}{J} - \frac{\gamma^2}{\kappa}} \; . \tag{2.15}$$

We now finally need the exact solution to the wave equation (2.10). The solution that is regular as $r \to 0$ is a particular Bessel function:

$$\beta(r) = K_0\left(\frac{\Omega}{r}\right) \; . \tag{2.16}$$

Expanding the Bessel function at infinity and performing a short computation we find

$$f(\Omega) = \frac{1}{\gamma^2 \log\left(\frac{\Omega}{\bar{r}_\star}\right)} \; , \qquad \bar{r}_\star \equiv 2 r_\star e^{-\Gamma_E} \; , \tag{2.17}$$

where $\Gamma_E$ is the Euler-Mascheroni constant. As claimed, the $r_\Lambda$ and $\kappa$ dependence has reassembled into a dependence only on the RG-invariant Landau pole scale $\bar{r}_\star$. Through (2.6) this determines the vacuum correlator; note that the presence of the Landau pole introduces logarithmic dependence on momenta and spoils conformal invariance, as anticipated.

## B. Finite temperature

We now consider this system at finite temperature. This is now in the universality class of the hydrodynamic theory studied in [7], except that the background magnetic field is zero; we note that a holographic study of thermodynamics and Kubo formulas with a nonzero magnetic field was recently performed in [5]. We will study the zero-field system at low frequencies and momenta and look for hydrodynamic modes.

We consider the system on a general black hole background of the form

$$ds^2 = g_{tt}(r)dt^2 + g_{rr}(r)dr^2 + g_{xx}(r)d\vec{x}^2 \tag{2.18}$$

We work in Lorentzian signature, so that $g_{tt}(r) < 0$. We assume that the metric has a finite-temperature horizon at $r = r_h$, so that $g_{tt}(r) \sim (r - r_h)$ and $g_{rr}(r) \sim (r - r_h)^{-1}$: we also assume that the metric is asymptotically AdS$_5$. The detailed form of the metric will not be important for our analysis, though for completeness we will sometimes specialize to the AdS$_5$-Schwarzschild metric:

$$ds^2 = r^2\left(-f(r)dt^2 + d\vec{x}^2\right) + \frac{dr^2}{r^2 f} \qquad f(r) = \left(1 - \frac{r_h^4}{r^4}\right) \tag{2.19}$$

where the temperature is related to the horizon radius $r_h$ by

$$T = \frac{r_h}{\pi} \ . \tag{2.20}$$

We begin by computing the analog of the charge susceptibility, i.e. in other words, we turn on a small constant source $b_{tx}$ and examine the response of $J^{tx}$, defining the susceptibility as the ratio of the response to the source. In the absence of any momentum, the equation of motion (2.5) is simply

$$\partial_r \left( \sqrt{-g} H^{rtx} \right) = 0 \qquad \sqrt{-g} H^{rtx} = -\gamma^2 \langle J^{tx} \rangle, \tag{2.21}$$

where the last equality follows from (2.4). Working in a gauge $B_{r\mu} = 0$ and imposing the usual horizon boundary condition $B_{t\mu}(r_h) = 0$, we easily find that

$$B_{tx}(r) = -\gamma^2 \langle J^{tx} \rangle \int_{r_h}^{r} dr' \frac{g_{rr} g_{tt} g_{xx}}{\sqrt{-g}} \tag{2.22}$$

The covariant form of the boundary condition (2.12) is

$$B_{\mu\nu}(r_\Lambda) - \gamma^2 \frac{J_{\mu\nu}}{\kappa} = b_{\mu\nu} \tag{2.23}$$

Using this to relate the field theory source $b_{tx}$ to the value of the bulk field $B_{tx}(r_\Lambda)$ we find that

$$\langle J^{tx} \rangle = \Xi b_{tx} \qquad \Xi^{-1} = \gamma^2 \left( \frac{1}{\kappa} - \int_{r_h}^{r_\Lambda} dr' \frac{g_{rr} g_{tt} g_{xx}}{\sqrt{-g}} \right), \tag{2.24}$$

where $\Xi$ is the susceptibility, for which we have now derived an explicit expression in terms of integrals over bulk metric coefficients.

Evaluating this on AdS$_5$ Schwarzschild we find

$$\Xi = \frac{1}{\gamma^2 \log \left( \frac{r_\star}{\pi T} \right)} \ . \tag{2.25}$$

As claimed, we see that $r_\Lambda$ and $\kappa$ have reassembled into the RG-invariant scale $r_\star$.

It is instructive to compare this result to the situation in free Maxwell gauge theory with gauge coupling $g$, in which case we find

$$\Xi_{\text{free}} = g^2 \ . \tag{2.26}$$

This matches nicely with the holographic result: we do not have a precisely marginal parameter $g$ in our computation, but we should instead interpret the logarithm appearing in (2.25) as measuring the running electromagnetic coupling at the scale of interest. Notice that in the regime of validity of the holographic regime $\gamma \ll 1$ implying we are exploring the strong coupling region in terms of the gauge coupling, as expected. On the other hand, the logarithmic running tames somewhat this growth in the IR.

We now compute the retarded finite-temperature correlators of the current $J^{\mu\nu}$. We study the correlators at finite frequency $\omega$ and spatial momentum $k$, orienting the spatial momentum in the $z$ direction. The correlation function can be decomposed into three channels by their transformation properties under the little group $SO(2)$ of rotations in the $xy$ plane:

1. Scalar: $\langle J^{tz} J^{tz} \rangle$. The conservation equation $\partial_\mu J^{\mu\nu} = 0$ sets this mode to zero, and we do not study it any further.

2. Vector: $\langle J^{ai} J^{bj} \rangle$ where $(a, b)$ run over $(t, z)$ and $(i, j)$ run over $(x, y)$. This channel is determined by a single scalar function. As we will see, it has a hydrodynamic diffusion mode.

3. Tensor: $\langle J^{xy} J^{xy} \rangle$. This channel contains the physics of Debye screening; however it has no hydrodynamic structure at low frequencies, and thus we will not study it any further in this work.

We therefore focus on the vector channel.

### 1. Hydrodynamics and diffusion

Using the techniques of [15], we can calculate the retarded correlator at small $\omega, k$ on any finite temperature metric. We define the current $J$ everywhere in the bulk as

$$J^{\mu\nu}(r) \equiv -\frac{1}{\gamma^2} \sqrt{-g} H^{r\mu\nu}(r) \tag{2.27}$$

When evaluated at the boundary, this reduces to the field theory current via (2.4). The bulk equations of motion can be conveniently written in terms of $J$ and $H$ as

$$\gamma^2 \partial_r J^{zx} + i\omega \sqrt{-g} g^{tt} g^{zz} g^{xx} H_{tzx} = 0 \tag{2.28}$$

$$-i\omega J^{tx} + ik J^{zx} = 0 \tag{2.29}$$

$$\partial_r H_{tzx} - \frac{\gamma^2}{\sqrt{-g}} \left( i\omega g_{zz} g_{xx} J^{zx} + ik g_{tt} g_{xx} J^{tx} \right) g_{rr} = 0 \tag{2.30}$$

where the first two are dynamical equations of motion and the last is the Bianchi identity. We now evaluate the ratio

$$\chi(r; \omega, k) \equiv \frac{J^{zx}(r)}{-H_{tzx}(r)} \tag{2.31}$$

as a function of the holographic coordinate $r$. As explained in [15], this is convenient as it takes a simple value at the horizon due to infalling boundary conditions.

$$\chi(r_h) = \Sigma(r_h) \qquad \Sigma(r) \equiv \frac{1}{\gamma^2} \sqrt{\frac{-g}{-g_{rr} g_{tt}}} g^{xx} g^{yy} \ . \tag{2.32}$$

On the other hand, when evaluated at the boundary it can be related to the field theory correlation function. Recall that the retarded correlator can be understood in linear response as the ratio between the response and the source:

$$\langle J^{\mu\nu}(\omega, k) \rangle = -G_{JJ}^{\mu\nu,\rho\sigma}(\omega, k) b_{\rho\sigma}(\omega, k) \ . \tag{2.33}$$

To understand the precise relationship between $G$ and $\chi$ we view (2.23) as an equation on the 4d boundary and take its 4d exterior derivative. Evaluating the $tzx$ component of the resulting equation, we find

$$H_{tzx}(r_\Lambda) - \frac{\gamma^2}{\kappa} \left( ik J^{tx} - i\omega J^{zx} \right) = db_{tzx} \tag{2.34}$$

Now using the current conservation equation (2.29) to eliminate $J^{tx}$ and considering a source where the only component turned on is $b_{zx}$, we find from (2.33) that

$$G_{JJ}^{zx,zx}(\omega, k) = \frac{-i\omega \chi(r_\Lambda; \omega, k)}{1 - \gamma^2 \frac{\chi(r_\Lambda; \omega, k)}{\kappa} \left( 1 - \frac{k^2}{\omega^2} \right) i\omega} \tag{2.35}$$

Finally, we now need to evolve $\chi(r)$ from the horizon at $r = r_h$ to the boundary at $r_\Lambda$. We thus use the bulk equations of motion (2.28) - (2.30) to obtain a flow equation for $\chi(r)$:

$$\partial_r \chi(r) = i\omega \sqrt{\frac{g_{rr}}{-g_{tt}}} \left( -\Sigma(r) + \frac{\chi^2}{\Sigma(r)} \left( 1 + \frac{k^2 g^{xx}}{\omega^2 g^{tt}} \right) \right) \tag{2.36}$$

In general this non-linear flow equation determines the full frequency dependence of the correlator and thus cannot be done analytically. However, it is very simple at low frequencies, and allows us to explicitly determine the hydrodynamic behavior in a manner that is independent of the details of the bulk background.

For example, if we assume a frequency and momentum scaling like $\omega \sim k^2$, then we can find the simpler equation

$$\frac{\partial_r \chi}{\chi^2} = -\frac{ik^2}{\omega} \frac{\sqrt{-g_{rr}g_{tt}}}{\Sigma} g^{xx} \tag{2.37}$$

which we may now integrate and insert into (2.35) to obtain the following expression for the correlator:

$$G_{JJ}^{zx,zx}(\omega, k) = \frac{-i\omega^2 \Sigma(r_h)}{\omega + iDk^2} \qquad D \equiv \Sigma(r_h) \left[ \int_{r_h}^{r_\Lambda} dr' \frac{\sqrt{-g_{rr}g_{tt}} g^{xx}}{\Sigma} + \frac{\gamma^2}{\kappa} \right] \tag{2.38}$$

In [7] it was shown that a universal definition of electrical resistivity $\rho$ in a dynamical $U(1)$ gauge theory is given by the Kubo formula:

$$\rho = \lim_{\omega \to 0} \frac{G^{xz,xz}(\omega, k=0)}{-i\omega} = \Sigma(r_h). \tag{2.39}$$

Note that the resistivity is given by an expression that depends only on horizon data; this can be thought of as a generalization of the usual holographic membrane paradigm [15] to higher-form currents. We see also that there is a hydrodynamic diffusion pole with a calculable diffusion constant $D$. From the expression for the charge susceptibility (2.24) we see that the diffusion constant satisfies an Einstein relation

$$\rho = \Xi D \ . \tag{2.40}$$

This is the diffusion of magnetic flux lines that are extended in the $x$ direction, modulated by a small momentum $k$ in the $z$ direction. The diffusive behavior of magnetic flux in a medium with a finite electrical conductivity is of course familiar from elementary electrodynamics: interestingly, here we see it arising in a strongly coupled medium. The existence of this diffusion mode follows from the zero-field limit of the hydrodynamic theory developed in [7], as we review in Appendix B.

To discuss the numerical values of the transport coefficients, we specialize to the AdS$_5$ Schwarzschild black hole. We find the resistivity and diffusion constant to be

$$\rho = \frac{1}{\gamma^2 \pi T} \qquad D = \frac{1}{\pi T} \log\left(\frac{r_\star}{\pi T}\right) \tag{2.41}$$

We may compare the resistivity to the (inverse) conductivity of the perturbative QED plasma with electromagnetic coupling $g$, computed in [16] to leading order in an expansion in inverse powers of $\log g$ to be:

$$\sigma^{-1} = \frac{g^2 \log g^{-1}}{CT} \tag{2.42}$$

where $C$ is a number related to the (electrically) charged particle content. While the gross temperature dependence is fixed by dimensional analysis, our holographic result for the resistivity does not have any logarithmic dependence on the Landau pole scale: one may interpret this as stating that it does not depend on the electromagnetic coupling, and thus our holographic result for the resistivity (unlike the thermodynamic result (2.25)) is significantly different from the perturbative result. This is a familiar theme in holography as the well known result for the shear viscosity of holographic theories makes manifest. Similar to that case, the resistivity scales with the number of degrees of freedom (charged under the 1-form symmetry).

### 2. Numerics and an emergent photon

It is not possible to go beyond the hydrodynamic limit analytically. It is however straightforward to obtain the spectral densities at arbitrary frequencies and momenta numerically.

Here we focus on one particular feature arising from such an investigation and illustrated in Figures 1 - 3.

At zero momentum the diffusion pole exhibited above sits at $\omega_{\mathrm{diff}} = 0$. Numerically we also see that at zero momentum there exists a purely overdamped pole[6]. Its precise location depends logarithmically on $r_\star$; at large $\frac{r_\star}{T}$ we find:

$$\frac{\omega_{\mathrm{osc}}}{2\pi T} \approx -i \frac{0.517}{\log\left(\frac{r_\star}{\pi T}\right)}, \tag{2.43}$$

where the dependence on $r_\star$ can be extracted analytically from the asymptotic structure of (2.35) at large $r_\Lambda$, but the prefactor we obtained numerically. This should be considered a heavily damped plasma oscillation; it depends on the model and cannot be obtained from hydrodynamics. Notice that at very low temperatures compared with the UV scale $r_\star$, $\log\left(\frac{r_\star}{\pi T}\right)$ becomes large and the pole approaches $\omega = 0$. As we have discussed the effective gauge coupling is given by $\Xi$ in this theory. At low temperatures it approaches a weakly coupled regime (although one must remember that $\gamma \ll 1$ for the holographic calculation to be trustworthy). One might expect charged matter to decouple in this regime and obtain a massless photon in the infrared. The behavior of the overdamped mode makes this plausible.

Let us now study the poles at finite momentum $k$. In the remainder of this section we fix $\frac{r_\star}{\pi T} = 1000$ when quoting all numerical values and plots.

As we illustrate in Figure 1, at small $k$ the diffusion pole moves straight down the imaginary axis following (2.38), and we observe numerically that the overdamped pole moves straight up the imaginary axis. As the theory is time-reversal invariant, any pole with a nonzero value of $\mathrm{Re}(\omega)$ must be accompanied by its time-reversal conjugate with $-\mathrm{Re}(\omega)$, and thus each isolated pole must remain on the imaginary axis.

---

[6] We focus here on two specific poles near the imaginary axis at $k = 0$: there also exist other non-hydrodynamic poles that we do not discuss.

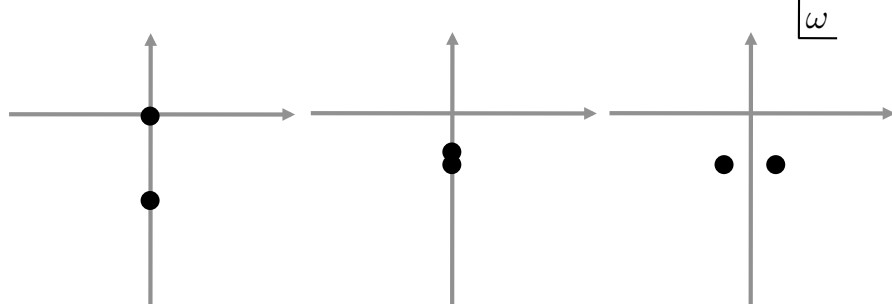

**FIG. 1:** Cartoon illustration of movement of diffusion and plasma oscillation pole in complex frequency plane as $k$ is increased from left to right; poles collide on imaginary axis at $\frac{k_\star}{2\pi T} \approx 0.037$ and then move off of axis symmetrically.

This remains true until the two poles collide at $\frac{k_\star}{2\pi T} \approx 0.037$, as seen in Figure 2. Each pole can now act as the time-reversal conjugate of the other, and indeed as we continue to increase $k$ we observe that $\mathrm{Im}(\omega)$ remains the same, but that the two poles symmetrically move off the imaginary axis, developing increasingly larger $\mathrm{Re}(\omega)$. Qualitatively similar behavior involving the merger of poles and subsequent movement off the imaginary axis has been seen in other holographic examples [17–20].

As we continue to increase $k$, eventually the dispersion relation approaches the relativistic $\omega \sim k$, as seen in Figure 3. In the language of conventional electrodynamics we would call this high-momentum mode the *photon*. Indeed, in the QED plasma we expect that at momenta much larger than the temperature, we expect the screening effects of the plasma to be unimportant, and thus the system should essentially behave as a free photon. Thus a linearly dispersing photon mode must somehow emerge from the hydrodynamic soup.

In this holographic model, there is no regime where the system is weakly coupled, but it nevertheless appears that a similar mechanism is at play, resulting in a gapless linearly dispersing mode. In particular, this hydrodynamic to collisionless (i.e. linearly dispersing) crossover is particularly sharp (happening precisely at $k_\star$), and the photon mode is actually continuously connected to the hydrodynamic diffusion mode. Note also that the initial pole (2.43) starts out closer to the origin at weaker electromagnetic coupling, and thus the crossover to the free photon regime happens faster (and the hydrodynamic regime is smaller) in this case.

It is an interesting question whether one can precisely interpret this emergent photon as a Goldstone boson of a generalized global symmetry at finite temperature [1, 6].

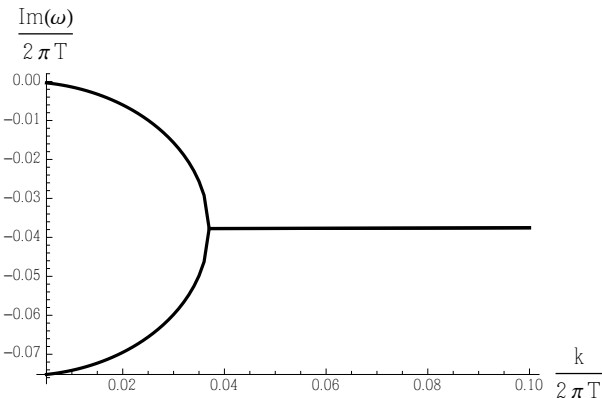

**FIG. 2:** Movement of imaginary part of diffusion pole (top) and damped plasma oscillation pole (bottom) as a function of momentum $k$. We have fixed $\frac{r_\star}{\pi T} = 1000$. Note merger at $\frac{k_\star}{2\pi T} \approx 0.037$.

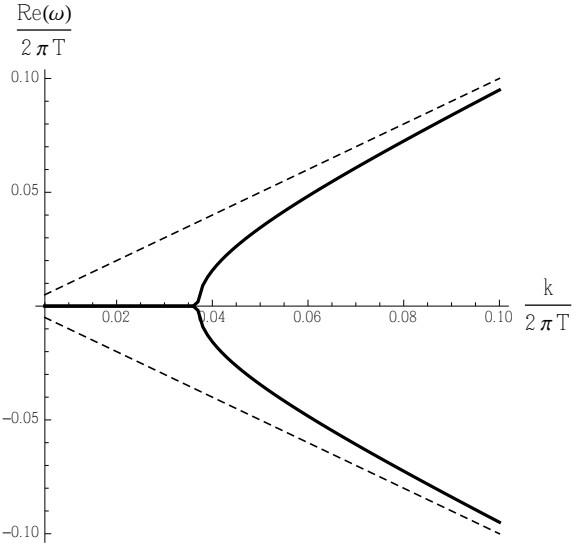

**FIG. 3:** Movement of real part of diffusion damped plasma oscillation poles as a function of momentum $k$, with $\frac{r_\star}{\pi T} = 1000$. For $k < k_\star$ both poles have zero real part; for $k > k_\star$ they symmetrically move off the imaginary axis, approaching a relativistic linear photon dispersion (dotted line) $\omega \sim k$ at large $k$.

## III.  CHERN-SIMONS TYPE HOLOGRAPHY

In this section we will study gauge potentials with Chern-Simons couplings in the bulk. The precise Chern-Simons couplings we will study will mix two different gauge potentials together and are often called BF theory. It turns out that such Chern-Simons theories in the bulk are naturally dual to *discrete* global symmetries on the boundary.

To understand this, we first note that continuous global symmetries in the boundary are dual to continuous gauge symmetries in the bulk. We are not aware of much study of discrete global symmetries in the context of holography; however from the reasoning above one might expect them to be dual to discrete *gauge* theories in the bulk. One simple way to understand this is to artificially construct a discrete global symmetry by breaking a continuous global symmetry; the bulk dual of this operation will result in a discrete gauge theory, as we will explain below.

Even away from this example, however, it is generally expected that there can exist no global (discrete or otherwise) symmetries in the bulk of a quantum gravity theory [21–23]. Thus the constraints arising from a field-theory discrete symmetry can only be encoded in a bulk discrete gauge theory. We will see that the machinery of this gauge theory is required to reproduce the boundary physics of a discrete global symmetry. The Chern-Simons theories that we will study are relevant precisely because they provide a continuum description of such discrete gauge theories [21, 24, 25].

The essential ideas here are the same both for 0-form and higher form symmetries, so we first work out the conventional 0-form case in detail. Here we begin with a system with a familiar continuous $U(1)$ symmetry and break it down to a $\mathbb{Z}_k$; we find that mixed Chern-Simons terms play an important role, and that the precise realization of the symmetry depends on boundary conditions. We then move on to the less-familiar case of a 1-form discrete symmetry.

## A. Discrete 0-form global symmetries and holography

Consider a 3d CFT with a continuous $U(1)$ 0-form global symmetry. We will call the microscopic current for this symmetry $j_e$. We consider also a scalar operator $\mathcal{O}$ that has charge $k > 1$ under it, and we imagine that there exists some other operator $\Psi$ with unit charge. If $\phi$ is the bulk field dual to $\mathcal{O}$, then the current sector of the CFT is represented by the following bulk action:

$$S = \int_{\mathcal{M}} \left( [(d - ikA)\phi] \wedge \star_4 \left[ (d + ikA)\phi^{\dagger} \right] - \frac{1}{4e^2} F \wedge \star_4 F + \cdots \right), \qquad (3.1)$$

where $\mathcal{M}$ represents a manifold that is asymptotically AdS$_4$. We will now break the $U(1)$ symmetry down to $\mathbb{Z}_k$. We will be careful to distinguish two cases:

1. Explicit breaking: we do this by adding a term $w \int d^4 x \, \mathcal{O}$ to the field theory action: in the bulk this corresponds to demanding that

$$\phi(r \to \infty) \sim w r^{\Delta - d} \qquad (3.2)$$

where $\Delta$ is the dimension of $\mathcal{O}$, i.e. we turn on the *large* falloff at infinity and specify its coefficient. We assume here that $\mathcal{O}$ is a relevant operator, $\Delta < d$, so that we retain control over the UV of the theory. Then in the infrared there should be no remnant of the original continuous $U(1)$ symmetry, and thus this theory should not have a conserved current associated to this charge.

2. Spontaneous breaking: here we imagine adding other couplings so that $\mathcal{O}$ develops a vev without a $U(1)$-breaking source added. This can be accomplished in a number of ways, e.g. adding a chemical potential as in holographic superfluids [26–28] or more simply via a symmetry-preserving double-trace coupling $\mathcal{O}^\dagger\mathcal{O}$ as in [29]. In this case we have $\phi(r \to \infty) \sim \langle\mathcal{O}\rangle r^{-\Delta}$, i.e the *small* falloff at infinity.

In both cases, however, the bulk field $\phi(r)$ develops a non-trivial profile. We denote its magnitude by $\rho(r)$, i.e. $\phi(r) = \rho(r)e^{i\theta(r)}$ with $\theta(r)$ a bulk Goldstone mode. The distinction between spontaneous and explicit breaking is contained in the form of $\rho(r)$ at large $r$.

The low-energy bulk action then becomes

$$S = \int_{\mathcal{M}} \left( \rho^2 (d\theta - kA) \wedge \star_4 (d\theta - kA) \right) + \cdots \tag{3.3}$$

We now dualize the Goldstone $\theta$ to a 2-form gauge field $B$ in the usual manner (see e.g. [21, 24]). We find after dualizing that

$$S = \int_{\mathcal{M}} \left( \frac{k}{2\pi} A \wedge dB - \frac{1}{4\rho^2} dB \wedge \star_4 dB \right) + \cdots \tag{3.4}$$

Here $B$ and $A$ have periods $\int_{\mathcal{M}_1} A = 2\pi\mathbb{Z}$, $\int_{\mathcal{M}_2} B = 2\pi\mathbb{Z}$ for all closed 1 and 2-cycles $\mathcal{M}_1$ and $\mathcal{M}_2$. The last term in the action is irrelevant from the point of view of the bulk, and we will ignore it from now on. Note this implies that the difference between spontaneous and explicit breaking will then be contained in the boundary conditions that we impose on the other fields at large $r$[7].

The key physics is in the first term: this Chern-Simons action describes a topological field theory in the bulk, defining a discrete $\mathbb{Z}_k$ gauge theory. There are no local degrees of freedom, as the equations of motion set both connections to be flat in the bulk:

$$dA = 0 \qquad dB = 0 . \tag{3.5}$$

---

[7] The fact that boundary conditions affect the symmetry algebra of the boundary theory is a well known fact in the context of the Chern-Simons formulation of $AdS_3$ gravity and its higher spin generalizations where the Drinfeld-Sokolov reduction is responsible from the reduction of the affine algebra $\hat{sl(N)}$ down to $W_N$.

The physical content of a $\mathbb{Z}_k$ gauge theory is instead in the braiding of massive excitations that are charged under the gauge fields. Here we have unit-charged particles that couple to the 1-form gauge field as $e^{i\int_C A}$ with $C$ a 1-dimensional worldline: these are excitations corresponding to quanta of the bulk field dual to a unit-charged operator in the boundary. This operator is *uncondensed* and thus its quanta must remain massive. On the other hand, recall that the charge-$k$ field $\phi$ in the bulk is condensed, and thus its quanta do not couple to $A$ as massive particles. We also have strings that couple to $B$ as $e^{i\int_W B}$, with $W$ a 2-dimensional worldsheet: in the original scalar representation of the theory (3.3) these are vortices that carry magnetic flux $\frac{2\pi}{k}$. The non-trivial braiding of particles and strings that is captured by the Chern-Simons term in (3.4) is just the Aharonov-Bohm phase of particles around flux tubes.

We now turn to the holographic interpretation of this bulk theory, which apparently should be dual to a boundary theory with either a global $U(1)$ symmetry spontaneously broken to $\mathbb{Z}_k$ or a theory with *only* a $\mathbb{Z}_k$ symmetry, depending on boundary conditions. We thus study the variation of the action (3.4) in the presence of a boundary.

We first study the case that corresponds to the spontaneously broken symmetry. In this case we should add boundary terms such that the total action is[8]

$$S_{tot} = \frac{k}{2\pi} \int_{\mathcal{M}} B \wedge dA + \frac{1}{2} \left( \frac{gk}{2\pi} \right)^2 \int_{\partial\mathcal{M}} B \wedge \star_3 B, \tag{3.6}$$

where $g$ is a free parameter that (as we will see) represents non-universal physics, and where we have picked its normalization to simplify subsequent equations. On-shell, the variation arises from a boundary term:

$$\delta S_{tot} = \frac{k}{2\pi} \int_{\partial\mathcal{M}} B \wedge \left( \delta A + \frac{g^2 k}{2\pi} \star_3 \delta B \right) \tag{3.7}$$

From this variational principle we conclude that we should take the field theory current $j$ and source $a$ to be

$$j_e = \frac{k}{2\pi} \star_3 B \big|_{\partial\mathcal{M}} \qquad a = - \left[ A + \frac{g^2 k}{2\pi} \star_3 B \right]_{\partial\mathcal{M}} \tag{3.8}$$

Finally, with the benefit of hindsight we define a 2-form $j_m$ that is the appropriately normalized Hodge dual of $j_e$:

$$j_m \equiv g^2 \star_3 j_e \tag{3.9}$$

This structure now captures all of the universal physics of a $U(1)$ symmetry spontaneously broken down to $\mathbb{Z}_k$:

---

[8] This boundary term is inspired by the 5 dimensional analog of this story discussed in detail in [30]. We will discuss their construction in more detail when we move to the case of 1-form global symmetries below.

1. *Conserved currents and Goldstone mode*: the bulk equations of motion (3.5)[9] imply the following equations for $j_e$ and $j_m$:

$$d \star_3 j_e = 0 \qquad d \star j_m = da \qquad (3.10)$$

We have a locally conserved 1-form current $j_e$, arising from the original spontaneously broken symmetry. We *also* have a 2-form current $j_m$ that is conserved up to a local function of the applied source: though this language is not usually used for a superfluid, this should be thought of as a mixed anomaly as in (1.11) or (1.13). Thus we have both a $U(1)$ 0-form and a 1-form symmetry: the 0-form symmetry is the original microscopic $U(1)$, but the 1-form symmetry is emergent in the infrared and measures vorticity. Here we will mostly focus on the realization of the 0-form symmetry.

The second equation implies that $j_e$ can locally be written as

$$j_e = \frac{1}{g^2} (d\psi - a) \qquad (3.11)$$

with $\psi$ a 0-form. Note that $g^{-2}$ plays the role of the superfluid stiffness. The conservation equation for $j_e$ then implies that

$$d \star_3 (d\psi - a) = 0, \qquad (3.12)$$

which is precisely the equation of motion for a $U(1)$ Goldstone mode $\psi$ in the presence of an external source $a$.

2. *Ward identities in the presence of charged operators*: we now consider adding charged objects in the bulk. We study a minimally electrically charged particle moving along a bulk curve $C$ that intersects the boundary at two points $x_1$ and $x_2$: this is holographically dual to an insertion of the unit-charged operator $\Psi$ and its conjugate at $x_1$ and $x_2$. We thus add a Wilson line term $\int_C A$ to the action to find that the equation of motion for $B$ and thus the conservation equation for $j$ is modified to read

$$\frac{k}{2\pi} dB = -\delta_1(C) \qquad d \star_3 j_e = \delta(x_1) - \delta(x_2) \qquad (3.13)$$

where $\delta_1(C)$ is a delta function along the curve $C$. The resulting non-conservation of $j$ is precisely the Ward identity for the current in the presence of the unit-charged operator $\Psi(x)$ at the points $x_1$ and $x_2$.

The other possible source is a 2-dimensional worldsheet coupling to $B$ in the bulk as $\int_W B$, intersecting the boundary along a 1-dimensional curve $\partial W$. In this case it is

---

[9] Note that if a bulk differential form vanishes, its projection down to the boundary also vanishes.

the equations of motion for $A$ and $d \star_3 j_m$ that are modified:

$$\frac{k}{2\pi} dA = -\delta_2(W) \qquad d \star_3 j_m = da - \frac{2\pi}{k} \delta_1(\partial W) \qquad (3.14)$$

In the field theory, the fact that $j_m$ has a source is usually interpreted as a unit-vorticity vortex along the curve $\partial W$. To understand this, first consider setting the external source $a$ to 0 and integrating $j_e$ over the boundary of an arbitrary large disc $D$ that includes $\partial W$: $\int_{\partial D} j_e = \int_D dj_e = -\frac{1}{g^2} \int_D d \star_3 j_m = \frac{2\pi}{g^2 k}$. We see that there is a net long-distance circulation of the microscopic current around $\partial W$, as we expect for a superfluid vortex. On the other hand, if we now turn on $a$, then this circulation can be stopped provided the net flux in $a$ is $\int_{\partial W} a = \frac{2\pi}{k}$, which is the expected flux quantization for a $\mathbb{Z}_k$ vortex. This is the Ward identity for the 1-form symmetry.

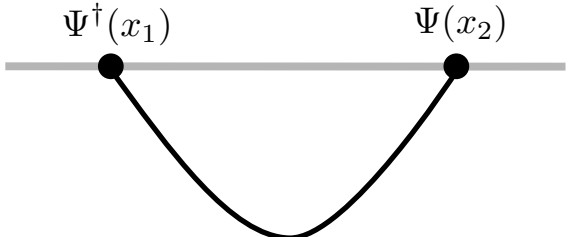

**FIG. 4:** Computing two-point function of $\Psi$ holographically: as the bulk charged worldline cannot break, the answer always depends strongly on separation between endpoints.

3. $\mathbb{Z}_k$ *order parameters*: finally, we look for order parameters for the broken symmetry. As the unit-charged operator $\Psi$ transforms under the unbroken $\mathbb{Z}_k$, it does not develop a vev and thus the two-point function $\langle \Psi^\dagger(x_1) \Psi(x_2) \rangle$ should vanish at large separation,

$$\lim_{|x_1 - x_2| \to \infty} \langle \Psi^\dagger(x_1) \Psi(x_2) \rangle = 0 \qquad (3.15)$$

As described above, this computation of this two-point function requires us to add a Wilson line $\int_C A$ to the action. By assumption the quanta of $\Psi$ are massive in the bulk and such a Wilson line will also come with a term $m \int_C ds$ that measures the length $L$ along the bulk worldline. A single Wilson line cannot break, and thus the geometric length grows with distance and will always suppress the correlator as $\exp(-mL)$ at large spacelike separation, as in Figure 4.

Consider now the operator[10] $\Psi^k$. This is invariant under the unbroken $\mathbb{Z}_k$ and so

---

[10] In the following discussion the properties of $\mathcal{O}$ should be completely analogous to those of $\Psi^k$. We choose

should develop a vev, i.e. the two-point function should saturate at large distances:

$$\lim_{|x_1-x_2|\to\infty}\langle(\Psi^k(x_1))^\dagger\Psi^k(x_2)\rangle = |\langle\Psi^k\rangle|^2 \tag{3.16}$$

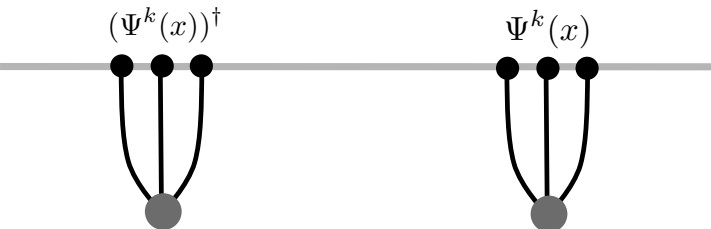

$(\Psi^k(x))^\dagger$           $\Psi^k(x)$

FIG. 5: Once $k$ Wilson lines can end on a bulk monopole, the 2-point function of widely separated insertions of $\Psi^k(x)$ will be dominated by configurations like this, where the answer is independent of separation.

In the bulk we now have $k$ Wilson lines. The key fact here is now that $k$ Wilson lines actually *can* end in the bulk, provided that they end on a *monopole event* in $B$, i.e at a bulk point $X$ where $d^2B(X)\neq 0$, or more properly where

$$\oint_{S^3(X)} dB = 2\pi, \tag{3.17}$$

where the integral is taken over a small $S^3$ surrounding $X$. To understand this, let us consider the bulk action with $k$ Wilson lines ending on a point $X$ which we excise from the manifold:

$$S = k\int_C A + \frac{k}{2\pi}\int_{\mathcal{M}} A\wedge dB \tag{3.18}$$

Now we perform a gauge transformation $A \to A + d\Lambda$ that vanishes at the boundary. The variation of the action receives, then, only contributions from the region around the monopole as

$$\delta_\Lambda S = k\Lambda(X) - \frac{k}{2\pi}\int_{S^3(X)} \Lambda dB \tag{3.19}$$

where the last term is the boundary term from the gauge variation of the Chern-Simons term. Thus we see that the termination of $k$ worldlines is consistent with gauge invariance provided that they end on a monopole in $B$. Once $k$ worldlines can break, the correlator will

---

to discuss $\Psi^k$ as it makes it manifest that it represents a source for $k$ Wilson lines associated to $\Psi$.

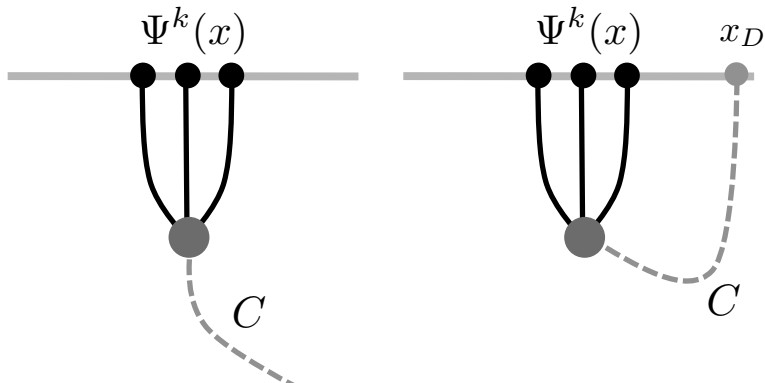

**FIG. 6:** When $k$ bulk Wilson lines can end on a monopole event in $B$ in the bulk, a Dirac string $C$ emerges from the monopole: we will distinguish the case where the string stays in the bulk *(left)* and the case where the string hits the boundary at $x_D$ *(right)*

eventually saturate at a value independent of separation as we expect for a broken symmetry, as we see in Figure 5.

We see that a very important role is played by the monopole events. In this case the monopole event is just an insertion of the original UV-complete field $\phi$ (as one can verify by tracing back through the duality and noting that the field $e^{i\theta}$ carries the correct monopole charge), but in general in quantum gravity we expect that such objects will always exist such that the charge lattice is filled (see e.g. [21–23]).

We now turn to the case of the *explicitly* broken symmetry. Our discussion mathematically parallels that of [31], though our interpretation is slightly different, as we focus on the role played by the conserved currents. To understand this, we first note that we have given physical importance to the 2-form gauge field $B$: from (3.8), it defines the field-theory current. We have also explained that we should allow monopole events in $B$. This combination may seem somewhat dangerous, as in the presence of a monopole, the field $B$ is not well-defined. Said differently, from the location of the monopole $X$ emerges a "Dirac string" (as shown in Figure 6) which in this case is a 1-dimensional worldline $C$ around which we have $\int_{S^2(C)} B = 2\pi$. In general Dirac strings are thought to be completely unobservable, as they have trivial braiding with any charged excitations and can be moved around by bulk gauge transformations.

In the presence of a boundary, however, this is not true. Indeed, the distinction between the spontaneous and explicitly broken symmetry depends on whether or not the Dirac string is allowed to intersect the boundary.

If the Dirac string is *not* allowed to intersect the boundary, then the discussion of the pre-

vious few paragraphs applies: $B$ evaluated at $\partial\mathcal{M}$ remains well-defined as it never intersects the Dirac string. There is no subtlety in the definition of $j$ (3.8) and thus the conserved $j_e$ implies that we have a continuous symmetry.

If the Dirac string *is* allowed to intersect the boundary, then the the charge defined as an integral on a 2-manifold $\mathcal{M}_2$

$$Q = \int_{\mathcal{M}_2} \star j_e \equiv \frac{k}{2\pi} \int_{\mathcal{M}_2} B \tag{3.20}$$

jumps discontinuously by $k$ as $\mathcal{M}_2$ is dragged across the Dirac string. Thus the intersection of the Dirac string with the boundary corresponds to the boundary insertion of a $k$-charged operator. In the presence of such insertions we no longer have a continuously conserved current, but the $\mathbb{Z}_k$ valued object

$$\mathbb{Q} = \exp\left(\frac{2\pi i}{k} Q\right) \tag{3.21}$$

still defines a conserved charge, in that its value does not change as $\mathcal{M}_2$ is moved through the (end of the) Dirac string, and the preserved symmetry is $\mathbb{Z}_k$.

Let us now examine whether the boundary conditions discussed above actually *allow* the Dirac string to intersect the boundary. Our discussion here will be mostly heuristic.

The boundary term in (3.6) associates an action cost to the existence of a Dirac string. Indeed, given the quantization condition $\int_{S^2} B = 2\pi\mathbb{Z}$ for a sphere surrounding the end of the string on the boundary we know the boundary term in the action (3.6) scales as:

$$S_{Dirac} \sim g^2 \Lambda \tag{3.22}$$

where we have only kept track of the $g$-dependence; $\Lambda$ is a UV cutoff, and the answer is UV divergent due to the divergence of the Goldstone mode near the core of the charge. The UV divergence indicates the configuration is not normalizable and, therefore, not allowed without the inclusion of further boundary terms that would cancel it. These terms would be directly responsible for the disappearance of the continuous symmetry. They are however not available to us in the effective low energy description (3.6). In the absence of these terms we conclude that for any finite $g$, UV divergences prohibit these Dirac strings, and we find a continuous $U(1)$ symmetry spontaneously broken down to $\mathbb{Z}_k$.

On the other hand, as $g \to 0$, Dirac strings are energetically allowed. Each intersection of the Dirac string with the boundary corresponds to the insertion of a charged field theory operator; thus these boundary conditions correspond in the field theory to having a non-trivial charged source turned on as in (3.2). From the point of view of (3.6) this is something of a singular point, as the boundary conditions degenerate to $A = 0$ at the boundary. The only information that remains in this theory is associated with topological objects such as

the charge operator (3.21) and Wilson lines (3.13). Such topological objects are the only universal information that we expect from the holographic representation of a $\mathbb{Z}_k$ discrete symmetry. In particular, the vortex configurations (3.14) violate the boundary conditions on $A$ and are no longer allowed.

We summarize:

1. Boundary conditions (3.6) with finite $g$ correspond to the case of a continuous $U(1)$ symmetry spontaneously broken down to $\mathbb{Z}_k$ with $g^{-1}$ corresponding to the superfluid stiffness for the associated gapless mode.

2. Boundary conditions (3.6) with $g = 0$ correspond to a field theory with *only* a $\mathbb{Z}_k$ symmetry, dual to a completely topological theory encoding the algebra of $\mathbb{Z}_k$ charges.

## B.   Discrete 1-form global symmetries and holography

In the previous section, we extensively discussed a mixed Chern-Simons term (3.4) in a four-dimensional bulk and showed that it represents the physics of a discrete $\mathbb{Z}_k$ 0-form symmetry in the holographically dual theory. We also showed that (depending on boundary conditions), it was possible that this $\mathbb{Z}_k$ was actually embedded into a spontaneously broken $U(1)$ symmetry.

Now we turn to the higher-form analog of this story: in other words, we introduce two 2-forms $B$ and $C$ and study the mixed Chern-Simons action

$$S_{CS}[B, C] = \frac{k}{2\pi} \int_{\mathcal{M}} C \wedge dB \tag{3.23}$$

where $B$ and $C$ are invariant under separate 1-form gauge invariances:

$$B \to B + d\Lambda \qquad C \to C + d\Gamma \tag{3.24}$$

where $\Lambda$ and $\Gamma$ are 1-forms. Invariance of the (exponential of the) action under large gauge transformations requires $k$ to be integer. The theory described by (3.23) defines a $\mathbb{Z}_k$ gauge theory in the bulk, describing the braiding statistics of string worldsheets that couple to $B$ and $C$. From the arguments in the previous section, we expect that this theory should be dual to a discrete $\mathbb{Z}_k$ symmetry that may (depending on boundary conditions) be embedded inside a spontaneously broken higher-form $U(1)$. A (mostly psychological) difference from the previous section is that it is not simple to fully UV complete this theory in the bulk: the analog of (3.1) is not straightforward. This is because the fundamental objects charged under the action of 1-form symmetries are not particles but extended objects. Thus our discussion will be purely in the Chern-Simons formulation.

We begin by noting that this theory has been extensively studied for a variety of purposes [30, 32–34]. A careful quantum treatment has been performed in [35]. Here we will essentially re-cast existing results in the language used in the previous section. We will discuss the results purely in terms of objects appearing in the low-energy action: however this precise term appears in Type IIB on $\text{AdS}_5 \times S^5$, and we will discuss the connection to $\mathcal{N} = 4$ SYM at the end.

We begin by studying the theory on a manifold with a four-dimensional boundary. We study the action with the following boundary terms [30, 36]:

$$S_{tot}[B, C] = S_{CS}[B, C] + \frac{1}{2}\left(\frac{gk}{2\pi}\right)^2 \int_{\partial M} C \wedge \star_4 C \qquad (3.25)$$

Here $g$ is a free parameter whose normalization we have picked so that it turns out to be the boundary Maxwell coupling. The bulk term in the variation is proportional to the equations of motion, which simply require that $B$ and $C$ be flat:

$$dB = 0 \qquad dC = 0 \ . \qquad (3.26)$$

The boundary variation is

$$\delta S_{tot}[B, C] = \frac{k}{2\pi} \int_{\partial M} C \wedge \left(\delta B + \frac{g^2 k}{2\pi} \star_4 \delta C\right) \qquad (3.27)$$

We can now identity the field theory current $J_e$ and 2-form source $b_e$:

$$J_e(x) = \left.\star_4 \frac{k}{2\pi} C\right|_{\partial M} \qquad b_e(x) = -\left.\left(B + \frac{g^2 k}{2\pi} \star_4 C\right)\right|_{\partial M}. \qquad (3.28)$$

$J_e$ is the microscopic $U(1)$ 2-form current; following (1.5), we also define a magnetic current as

$$J_m \equiv g^2 \star_4 J_e \qquad (3.29)$$

We may now study all of the same considerations as in the previous section:

1. *Conserved currents and Goldstone photon:* We begin with a discussion of the local conservation equations. From the bulk flatness conditions (3.26) and the definition of the source (3.28) we find

$$d \star_4 J_e = 0 \qquad d \star_4 J_m = db_e \qquad (3.30)$$

   These conservation laws are the higher-form analog of (3.10), describing two conserved 2-form currents $J_e$ and $J_m$. From the point of view taken here, $J_m$ is an emergent symmetry. These are also precisely equivalent to the conservation laws derived from the action for a free $U(1)$ Maxwell gauge field (1.11). In (3.10) we showed that the

current could be written in terms of the action of a free Goldstone: here we can follow precisely the same logic to conclude that

$$J_e = \frac{1}{g^2} (dA_e - b_e) \qquad d \star_4 (dA_e - b_e) = 0 \qquad (3.31)$$

with $A_e$ an arbitrary 1-form that is the higher-form Goldstone mode. It is also the Maxwell photon. Thus the Maxwell photon can be thought of as the Goldstone mode of a spontaneously broken 1-form symmetry [1].

2. *Ward identities in the presence of charged operators:* The two possible charged operators are string worldsheets $W$ that couple to $B$ and $C$ respectively. For strings that couple to $B$ we add a term $\int_W B$ to the action to find

$$\frac{k}{2\pi} dC = -\delta_2(W) \qquad d \star_4 J_e = \delta_1(\partial W) \qquad (3.32)$$

where $\partial W$ is the one-dimensional intersection of the string worldsheet with the boundary: we see that in the field theory this boundary represents a line-like operator $L_B(\partial W)$ that is charged under the electric symmetry $J_e$.

We now add a string that couples to $C$: we find

$$\frac{k}{2\pi} dB = -\delta_2(W) \qquad d \star_4 J_m = \left( db_e - \frac{2\pi}{k} \delta_1(\partial W) \right) \qquad (3.33)$$

Thus the line-like operator $L_C(\partial W)$ living on $\partial W$ is charged under $J_m$. From the point of view of the boundary photon, $\partial W$ is the worldline of a magnetic monopole. Note that it appears to have fractional charge; we will return to this point later.

3. $\mathbb{Z}_k$ *order parameters:* We first recall what it means for a line-like operator to be "condensed": as described in [1], we say that a line-like operator is condensed if it obeys a perimeter law, as this is the higher-form generalization of the factorization of local operators (3.16). Any dependence on the geometric data characterizing the loop that is stronger than this (e.g. an area law) is the analog of the uncondensed result (3.15).

Now consider a single bulk string coupled to $B$, dual to the insertion of the line operator $L_B(\partial W)$. As the string cannot end in the bulk, its bulk tension will result in an expectation value $\langle L_B(\partial W) \rangle$ that depends on its radius more strongly than a perimeter law (though, depending on the IR geometry, perhaps less strongly than an area law). Thus we say that $L_B(\partial W)$ is uncondensed.

We now turn to $L_B(\partial W)^k$, dual to $k$ bulk strings. Following arguments precisely analogous to those around (3.18), $k$ bulk strings can end on a monopole in $C$, i.e. on

a one-dimensional worldline $P$ around which we can integrate $dC$:

$$\int_{S^3(P)} dC = 2\pi \tag{3.34}$$

This monopole will generally have mass, but the tension in the strings will generically pull this object towards the boundary, effectively localizing the worldsheet of the $B$-string near the boundary and resulting in a true perimeter law for $\langle L_B(\partial W)^k \rangle$. Thus $L_B(\partial W)^k$ is condensed and we see the $U(1)$ symmetry generated by $J$ is spontaneously broken to $\mathbb{Z}_k$.

Precisely the same arguments apply for $k$ copies of the object charged under $C$, where $k$ $C$-strings are allowed to end on a monopole in $B$. Thus if all monopole events are allowed, the symmetry generated by $\star_4 J$ is also spontaneously broken down to $\mathbb{Z}_k$.

We now turn to the issue of spontaneous versus explicit breaking of the $U(1)'s$: again, our discussion parallels that around the lower-form case. The monopole in $C$ has a Dirac string (which is a 2-dimensional object around which $\int_{S^2} C = 2\pi$): if this $C$-type Dirac string is allowed to intersect the boundary then the current $J_e$ ceases to be well-defined and we can only consider the exponential of the integrated charge

$$\mathbb{Q}_e \equiv \exp\left(\frac{2\pi i}{k} \int_{\mathcal{M}_2} \star_4 J_e\right) \tag{3.35}$$

which is only defined modulo $k$, breaking the symmetry down to $\mathbb{Z}_k$.

Similarly, the monopole in $B$ can have a Dirac string: here the situation is slightly different, as we see from (3.28) that actually the definition of the source $b$ itself has become ill defined. If we assume that the source is trivial, then we can conclude that the gauge-invariant charge is again the $\mathbb{Z}_k$-valued object

$$\mathbb{Q}_m \equiv \exp\left(\frac{i}{k} \int_{\mathcal{M}_2} \star_4 J_m\right) \tag{3.36}$$

As discussed around (3.22), the energetics of the intersection of the Dirac strings with the boundary depends on the value of $g$. A computation paralleling (3.22) shows that the boundary action of a $C$-type Dirac string (i.e. a boundary charge for $J_e$) scales as $g^2 \Lambda L$ and that of a $B$-type Dirac string (i.e. a boundary charge for $J_m$) scales as $g^{-2} \Lambda L$, with $\Lambda$ the UV cutoff and $L$ the boundary length of the intersection. Thus we conclude that

1. For finite $g$, neither type of Dirac string is permitted: the boundary symmetry is $U(1)_e \times U(1)_m$ spontaneously broken down to $\mathbb{Z}_k \times \mathbb{Z}_k$, and $g$ is the gauge coupling of the boundary photon.

2. For $g = 0$, $C$-type Dirac strings are permitted. This breaks $U(1)_e$ explicitly down to $\mathbb{Z}_k$: note that the boundary conditions become simply $B = 0$, which prohibits the magnetic charges (3.33) and thus we have only the single electric $\mathbb{Z}_k$.

3. For $g = \infty$, $B$-type Dirac strings are permitted, breaking $U(1)_m$ explicitly down to $\mathbb{Z}_k$. The boundary conditions are now $C = 0$, prohibiting the electric charges (3.32) and leaving only the single magnetic $\mathbb{Z}_k$.

Up till now, our discussion has been purely in terms of the objects appearing in the low-energy action. We now discuss the connection with $SU(N)$ $\mathcal{N} = 4$ Super-Yang-Mills. In particular, since the early days of AdS/CFT, it has been known that the usual action of Type IIB string theory compactified on $\mathrm{AdS}_5 \times S^5$ has precisely such a Chern-Simons term, with $k = N$, $B$ being the NS-NS 2-form, and $C$ the R-R 2-form [32, 33, 37]. Thus our results may immediately be taken over. The objects coupling to $B$ are fundamental strings, and those coupling to $C$ are D1 branes; they are dual respectively to Wilson and t'Hooft lines on the boundary. Monopoles in $B$ are $D5$ branes wrapped on the $S^5$ (i.e. Witten's baryon vertex [38]), and monopoles in $C$ are $NS5$ branes wrapped on the $S^5$.

Note now that the global form of the gauge group determines the spectrum of allowed line operators [39–41] and thus is relevant for the structure of generalized symmetries [1, 42]. The three cases above seem to realize $U(N)$, $SU(N)$, and $SU(N)/\mathbb{Z}_N$, as we now discuss[11].

Case 1 corresponds to $U(N)$ gauge theory. The full generalized symmetry group of $U(N)$ gauge theory is $U(1)_e \times U(1)_m$. As we briefly review in Appendix C, in $U(N)$ gauge theory one is allowed both Wilson and t'Hooft lines. From the point of view of the continuous $U(1)$ generalized symmetry currents, minimally charged t'Hooft lines appear to have magnetic charge that is $1/N$-th the minimum $U(1)$ Dirac quantum; this is precisely what we see in (3.33). The "singleton" boundary photon identified above can be thought of as the $U(1)$ factor of the $U(N)$ gauge group; as expected, it lives on the boundary and does not interact with the bulk except through charged objects.

Case 2 corresponds to $SU(N)$ gauge theory, where we have $\mathbb{Z}_N$ Wilson lines (i.e. fundamental strings coupled to $B$) but t'Hooft lines are not allowed.

Case 3 corresponds to $SU(N)/\mathbb{Z}_N$ gauge theory, where we have $\mathbb{Z}_N$ t'Hooft lines (i.e. D1 strings coupled to $C$) but Wilson lines are not allowed.

As far as we understand the precise classification above is novel but is broadly consistent with the existing literature on this subject. It would be instructive to subject this picture to more detailed tests.

---

[11] We are very grateful to D. Tong for instructive discussions about the contents of this section.

## IV.  MAXWELL-CHERN-SIMONS TYPE HOLOGRAPHY

In the first part of this paper we considered the Maxwell term alone for a 2-form gauge field, and in the second we considered a mixed Chern-Simons term for two 2-form gauge fields. We now consider combining these ingredients by studying both together, i.e. we study the action

$$S = \int d^5x \sqrt{-g} \left( -\frac{1}{6\gamma^2} \sqrt{g} H_{MNP} H^{MNP} - \frac{1}{6\gamma'^2} G_{MNP} G^{MNP} + \frac{k}{24\pi} \epsilon^{MNPQR} B_{MN} G_{PQR} \right) \tag{4.1}$$

where $H = dB$ and $G = dC$.

Note that this is the most general quadratic action for the fields $(B, C)$. However, from the point of view of the bulk, the Maxwell terms are irrelevant perturbations to the long-distance physics described by the Chern-Simons term. In addition to the physics of flat connections described in the previous section, we will now have an extra topologically massive mode for the gauge fields [43]. Similar topologically massive bulk gauge fields have been studied in the context of $AdS_3/CFT_2$ in [44, 45].

Note that we may apparently remove a parameter from the problem by rescaling $C$ to obtain the same normalization for the two bulk Maxwell terms

$$S = \frac{1}{\gamma^2} \int d^5x \sqrt{-g} \left( -\frac{1}{6} H_{MNP} H^{MNP} - \frac{1}{6} G_{MNP} G^{MNP} + \lambda \epsilon^{MNPQR} B_{MN} G_{PQR} \right) \tag{4.2}$$

where $\lambda \equiv \frac{k\gamma'\gamma}{24\pi}$. The quantum physics still depends on $\gamma'$ as the rescaling modifies the quantization conditions on the periods of $C$ [44]. However in this section our considerations will be purely classical in the bulk, and we can express all of our results in terms of $\lambda$ and $\gamma$.

### A.  Operator content

We begin by describing the operator content of the dual theory. Consider first setting the coefficient of the Chern-Simons mixing term $\lambda$ to 0. This results in two copies of the theory studied in Section II, which will have two decoupled boundary currents, each of dimension 2. If we now turn on the Chern-Simons coupling, the IR structure of the bulk theory is strongly modified: as described in detail in Section III B, the flat part of both $(B, C)$ is now dual to a single tensor operator $J$. As the number of degrees of freedom should remain the same in the presence of the mixing term[12], the non-flat part of $(B, C)$ must contribute

---

[12] An interesting subtlety is that in the theory with no mixing term, we have two separately conserved currents. However in the theory with the mixing term, we have a single conserved current $J$ that obeys

another tensor operator. This is dual to a massive mode in the bulk and has a non-trivial scaling dimension that we will now determine.

The equations of motion are

$$\nabla_M G^{MNP} - \lambda \epsilon^{ABCNP} H_{ABC} = 0 \ , \tag{4.3}$$

$$\nabla_M H^{MNP} + \lambda \epsilon^{ABCNP} G_{ABC} = 0 \ . \tag{4.4}$$

It is often convenient to assemble these into a single complex 2-form $Z$ and its field strength $W$:

$$Z \equiv B + iC \qquad W \equiv dZ = H + iG \tag{4.5}$$

in which case the two equations of motion can be combined into one, which we write in form notation as

$$d \star_5 W = -6\lambda i W \ . \tag{4.6}$$

Following a treatment of a lower-dimensional problem in [45], we would now like to separate $Z$ into a flat part $Z_0$ and a non-flat part $\zeta$:

$$Z = Z_0 + \zeta \qquad dZ_0 = 0 \tag{4.7}$$

where $Z_0$ is flat and presumably $\zeta$ contains the massive mode that we are interested in. Of course this split is ambiguous, as we can always transfer more flat parts of the connection into $\zeta$. To fix this ambiguity, we first note that $\zeta$ satisfies the equation

$$d \star_5 d\zeta = -6\lambda i d\zeta \ . \tag{4.8}$$

We may now *choose* $\zeta$ such that it satisfies the following equation:

$$\zeta = \frac{i}{6\lambda} \star_5 d\zeta \tag{4.9}$$

(4.9) implies that (4.8) is satisfied: it is however not the most general solution to (4.8), and the choice of this particular $\zeta$ amounts to a particular division of the connection into flat and non-flat pieces.

The physics stored in the flat part $Z_0$ was described in the previous section: we would now like to study the physics in $\zeta$. To that end, we study linearized perturbations around Lorentzian AdS$_5$, written as

$$ds^2 = \frac{dr^2}{r^2} + r^2 \left( -dt^2 + dx^2 + dy^2 + dz^2 \right) \ . \tag{4.10}$$

---

two separate conservation equations (for $J$ and $\star_4 J$), and another higher-dimension operator that obeys no conservation law at all: thus the number of independent components is preserved though the constraints are redistributed.

We would first like to determine the conformal dimensions; thus we study solutions to (4.9) that are independent of the field theory directions. Notice that this implies $\zeta_{r\mu} = 0$ as our solution satisfies $d\zeta|_{\text{boundary}} = 0$. $\zeta$ is, therefore, a 2-form with components only in the field theory directions. The first order equation (4.9) becomes:

$$r\partial_r\zeta = 6i\lambda \star_4 \zeta \tag{4.11}$$

where $\star_4$ is the 4d Hodge star with respect to the flat Lorentzian metric $ds^2 = -dt^2 + dx^i dx^i$. It is now useful to introduce the projectors onto self-dual and anti-self-dual 2-forms in 4d:

$$P_\pm \equiv \frac{1}{2}\left(1 \pm i\star_4\right) \qquad i\star_4 P_\pm = \pm P_\pm \qquad P_\pm^2 = P_\pm \qquad P_+P_- = 0 \tag{4.12}$$

Defining a basis of definite chirality boundary 2-forms using $\zeta_\pm = P_\pm\zeta$, we see that (4.11) becomes

$$r\partial_r\zeta_\pm = \pm 6\lambda\zeta_\pm \tag{4.13}$$

and thus the general solution takes the form

$$\zeta(r) = r^{6\lambda}\zeta_+ + r^{-6\lambda}\zeta_- \tag{4.14}$$

Thus we see the expected two falloffs at infinity, where the corresponding polarization tensors obey a certain projection condition. This is the usual structure at infinity for a first-order dynamical system in AdS/CFT (see e.g. the well-studied case of fermions [46, 47]).

Via the usual rules we expect that if $\lambda > 0$ then $\zeta_+$ is the source and $\zeta_-$ is the response. Note that as $\lambda \to 0$, the two solutions coincide and we obtain the logarithm seen in (2.11). To find the dimension $\Delta$ of the dual operator, we note that regardless of the spin of the operator, the difference between the two exponents is always equal to the difference between $\Delta$ and $4 - \Delta$, which means that

$$\Delta = 2 + 6|\lambda| \tag{4.15}$$

The dimension is always given by the expression above, though the choice of which of the two falloffs is normalizable depends on the sign of $\lambda$ so that we remain above the unitarity bound for a conserved current.

The existence of this operator is somewhat interesting: as it arises from the quadratic part of the bulk action, it is a generic feature of any holographic theory. We also note that in the dual to a large $N$ gauge theory, $\lambda \sim N^{-1}$ [33] and thus the dimension is very close to that of a conserved current. It would be interesting to understand if this operator has a clean interpretation in the dual theory.

Finally, we note that our treatment is incomplete: technically speaking, a careful identification of sources and vevs requires that we holographically renormalize the theory defined by (4.2). We leave such an analysis for future study.

## B. Backreacted scaling solutions

In this section we couple the above system to gravity and demonstrate the existence of new anisotropic scaling solutions. We study stationary points of the following action:

$$S = \int d^5 x \sqrt{-g} \left[ \frac{1}{2\kappa^2}(R+12) + \frac{1}{\gamma^2} \left( -\frac{1}{6}H_{MNP}H^{MNP} - \frac{1}{6}G_{MNP}G^{MNP} + \lambda \epsilon^{MNPQR}B_{MN}G_{PQR} \right) \right],$$
(4.16)

where we are working in units where this system admits an $AdS_5$ vacuum with unit AdS radius. The full equations of motion are those for the gauge fields (4.3) and (4.4), together with those arising from varying the metric:

$$\frac{1}{2\kappa^2}\left( R_{AB} - \frac{1}{2}g_{AB}R - 6g_{AB} \right) + \frac{1}{6\gamma^2}\left( \frac{g_{AB}(H^2+G^2)}{2} - 3H_{ANP}H_B{}^{NP} - 3G_{ANP}G_B{}^{NP} \right) = 0$$
(4.17)

Note that the Chern-Simons term does not directly contribute to the gravitational equations of motion as it is topological: however, as it affects the dynamical equations for the gauge fields it dramatically changes the character of the allowed gravitational solutions.

We first briefly discuss known solutions when $\lambda = 0$. In this case we have two decoupled 2-form gauge fields coupled to gravity. In the 5d bulk these 2-forms can be dualized to 1-form vector fields, and we are thus simply discussing solutions to the very well-studied Einstein-Maxwell theory in $AdS_5$ in a different bulk duality frame. If these solutions carry electric charge, then we have the well-known AdS-Reissner-Nordstrom black branes [48], which have $AdS_2$ IR asymptotics at zero temperature (see e.g. [49, 50] for reviews). On the other hand, if they have a nonzero magnetic field along (say) the $x$ direction, then an asymptotically $AdS_5$ solution is not analytically known, but there exists an exact IR scaling solution that is $AdS_3 \times \mathbb{R}^2$, where the $AdS_3$ is made out of $(t, r, x)$ [51, 52]. Returning to the duality frame used in this paper, such solutions correspond to having a nonzero boundary $J^{tx}$ and have been studied from the point of view of generalized symmetries and magnetohydrodynamics in [5].

We now return to finite $\lambda$. Somewhat surprisingly, we can still find exact scaling solutions to the backreacted system, though we have not been able to analytically construct a full bulk RG flow to $AdS_5$ in the UV. We expect that such RG flows could be found numerically.

The IR solution is a product of $AdS_3$ and a shrinking $\mathbb{R}^2$. It is similar to a Lifshitz geometry [53], in that the dimensions $(t, x)$ scale at a different rate from $(y, z)$. From this perspective they represent the emergence of a (deformed) $CFT_2$ in the IR living on the worldsheet of magnetic flux tubes in the boundary $CFT$. The solution is:

$$B = \frac{b_0}{u^2}dt \wedge dx \quad C = \frac{c_0}{u^{\frac{2}{\xi}}}dy \wedge dz \quad ds^2 = L^2\frac{du^2 - dt^2 + dx^2}{u^2} + \frac{1}{u^{\frac{2}{\xi}}}\left( dy^2 + dz^2 \right), \quad (4.18)$$

which is invariant under the scaling isometry

$$u \to \sigma u \qquad t \to \sigma t \qquad x \to \sigma x \qquad y \to \sigma^{\frac{1}{\xi}} y \qquad z \to \sigma^{\frac{1}{\xi}} z \qquad (4.19)$$

and thus $\xi$ plays a role analogous to the Lifshitz dynamical exponent $z$. As usual for scaling geometries, the solution is unique in that the parameters appearing in the solution are completely fixed in terms of bulk coupling constants. A solution of the equations of motion is found provided that

$$L = \sqrt{\frac{2}{3}} \sqrt{\frac{1}{1 - 3\lambda^2 + \sqrt{1 - 6\lambda^2 - 27\lambda^4}}} \qquad (4.20)$$

$$b_0 = \frac{L^2}{2\kappa} \sqrt{2 - 3L^2 + 81L^4\lambda^4} \qquad (4.21)$$

$$c_0 = \frac{b_0}{3\lambda L^3} \qquad (4.22)$$

$$\xi = (3L\lambda)^{-2} \qquad (4.23)$$

If we imagine taking $\lambda \to 0$, then we see that $L \to \frac{1}{\sqrt{3}}$ and $\xi \to \infty$: the $C$ field becomes pure gauge and decouples, and the $(y, z)$ directions cease to shrink. The bulk geometry becomes the magnetic brane AdS$_3 \times \mathbb{R}^2$ of [51].

For nonzero $\lambda$ this is a novel solution. We note also that at $\lambda = \frac{1}{3}$ these solutions become once again $AdS_5$ backgrounds: $\xi \to 1$ in this limit as well as $L \to 1$. The gauge fields turn off and we recover the purely gravitational solution. Beyond this point solutions cease to exist, as they can no longer be supported by fluxes. While it is hard to interpret this fact without knowing the exact interpolating solutions from $AdS_5$ to the IR, it is interesting to notice that the solutions exactly disappear when the massive mode found in (4.14) becomes dual to marginal boundary operators (4.15). One might expect that once that value is crossed no deformation caused by such operator can affect the IR, so new scaling solutions would not be available at $\lambda > 1/3$.

While we have not done so here, this IR scaling solution can in principle be connected to an asymptotically AdS$_5$ solution, and the resulting spacetime is dual to a particular state of $\mathcal{N} = 4$ SYM, presumably corresponding to color flux tubes oriented in the $x$ direction. It would be very interesting to understand the physics described here from the field-theoretical point of view.

## V. CONCLUSION

In this work we have studied various aspects of generalized global symmetries in quantum field theories with holographic duals, focusing on 1-form symmetries in four-dimensional

quantum field theories. We briefly summarize the main points of our analysis below.

We began with a study of a single continuous conserved 2-form current $J$, dual to an antisymmetric tensor field with a Maxwell action in a five-dimensional bulk. We showed that this field theory is not conformal: instead the double-trace coupling $J^2$ runs logarithmically and the theory has a Landau pole in the UV. We further studied this theory at finite temperature, computing transport coefficients and showing the existence of a diffusion mode that is compatible with the hydrodynamic analysis of [7].

We then turned to a study of discrete symmetries. We began with the case of a 0-form discrete symmetry: this is just a conventional discrete symmetry in field theory (i.e. where the charge is defined on a codimension-1-manifold, or a "time-slice"). We argued that in general, a field-theoretical discrete symmetry is holographically dual to a discrete gauge theory in the bulk. It is well-known that such gauge theories have a low-energy description in terms of a mixed Chern-Simons theory, and we explained in detail how to understand the universal physics of the discrete symmetry from the bulk topological theory. We also showed that this discrete symmetry may be embedded inside a continuous symmetry which can be spontaneously broken; in the Chern-Simons description, the associated Goldstone boson can be thought to live on the boundary. The distinction between explicit and spontaneous breaking arises from different boundary conditions on the Chern-Simons gauge fields.

Next, we studied a 1-form discrete symmetry, which has a similar Chern-Simons description in terms of 2-form antisymmetric tensor gauge fields. This case is relevant for the study of $\mathcal{N} = 4$ super-Yang-Mills theory, which is expected to realize (at least) a discrete higher form symmetry. The precise symmetry structure and spectrum of charged line operators depends on the precise presentation of the gauge group: in particular, we clarify the distinction between the holographic duals of the $U(N)$, $SU(N)$, and $SU(N)/\mathbb{Z}_N$ gauge theories and explain the holographic boundary conditions that realize the generalized symmetry structure expected for the three different cases. In the $U(N)$ case there is a continuous Abelian global 1-form symmetry that is spontaneously broken down to a discrete subgroup: we identify the boundary photon (i.e. the "U(1)") as the Goldstone mode of the symmetry breaking.

Finally, we studied the bulk theory with both the Maxwell and Chern-Simons terms for the 2-form gauge fields. Here the higher-derivative Maxwell terms result in new massive modes in the bulk which are dual to higher-dimension tensor operators in the boundary. We perform a preliminary analysis of this theory, computing the dimension of the new operator. We also study gravitationally backreacted solutions to this theory, finding an exact IR scaling solution that appears to be dual to color flux tubes extended in one of the spatial directions.

There are many directions for future research. We expect that the detailed understanding of the implementation of discrete symmetries (both conventional and higher-form) in AdS/CFT will have holographic applications. In particular, it would be interesting to un-

derstand if the tools of holography can be helpful in recent efforts to understand topological phases of matter (see e.g. [54]) and the phase structure of non-Abelian gauge theories from the point of view of generalized symmetry. More concretely, the existence of the higher-dimension tensor operator alluded to above is somewhat mysterious from the field-theoretical point of view. As it arises from the most general possible quadratic action in the bulk, we expect it to have an interpretation in the field theory. It would also be interesting to connect the scaling solution found above to an asymptotically AdS$_5$ solution and interpret it from the point of view of color flux tubes in gauge theory.

Finally, the analyses (both holographic and otherwise) performed here indicate an interesting structure involving the interplay between conformality, spontaneously broken generalized $p$-form symmetry, and emergent $d - p - 2$ form symmetry. While we will comment further on some of these issues in [6], we expect that there is still much to learn, and that further study of generalized symmetries will teach us much about the structure of field theory.

### Acknowledgements

We thank N. Bobev, B. Craps, S. Cremonesi, X. Dong, S. Hartnoll, E. Katz, P. Koroteev, A. Potter, N. Poovuttikul, S. Ross, E. Shaghoulian and D. Tong for illuminating discussions, and S. Grozdanov for collaboration on related issues and for sharing [5] prior to publication. NI would like to thank Delta ITP at the University of Amsterdam and the Aspen Center for Physics (which is supported by National Science Foundation grant PHY-1607611) for hospitality during the completion of this work. NI is supported in part by the STFC under consolidated grant ST/L000407/1. This work is part of the Delta ITP consortium, a program of the NWO that is funded by the Dutch Ministry of Education, Culture and Science (OCW).

### Appendix A: Conventions and differential form identities

In this work we normally use $M, N$ to refer to 5d bulk indices, $\mu, \nu$ to refer to 4d field theory bulk indices, and $i, j$ to refer to 3d spatial indices. Section III A involves a 4d bulk and a 3d boundary; however we write that section entirely using index-free differential forms.

Our our conventions for differential forms are those of [55], and we record some useful identities below:

$$d(\omega_p \wedge \eta_q) = d\omega_p \wedge \eta_q + (-1)^p \omega_p \wedge d\eta_q \tag{A1}$$

$$\omega_p \wedge \eta_q = (-1)^{pq} \eta_q \wedge \omega_p \tag{A2}$$

$$\omega_p \wedge \star \eta_p = \eta_p \wedge \star \omega_p \tag{A3}$$

The square of the Hodge star acting on a $p$ form in $n$ dimensions on a metric with $s$ minus signs in its eigenvalues is

$$\star^2 = (-1)^{s+p(n-p)} . \tag{A4}$$

In particular, in Lorentzian signature in 4d acting on a 2-form, we have $\star_4^2 = -1$.

Sadly, this subject involves many factors of $2\pi$. We pick conventions where electric and magnetic charges satisfying Dirac quantization satisfy $Q_e \equiv \int \star_4 J_e = \mathbb{Z}$ but $Q_m \equiv \int \star_4 J_m = 2\pi\mathbb{Z}$.

### Appendix B: MHD diffusion mode at zero magnetic field

A theory of magnetohydrodynamics from the point of view of generalized symmetries was developed in [7]. Here we specialize that theory to the case with zero background magnetic field, ending with the derivation of the diffusion mode obtained holographically in (2.38). If the background field is zero, then the fluctuations of the 2-form current $J^{\mu\nu}$ and the stress tensor decouple, and we thus consider only $J^{\mu\nu}$.

In ideal hydrodynamics we have

$$J_{(0)}^{\mu\nu} = 2\rho u^{[\mu} h^{\nu]} \qquad h^2 = 1 \qquad u^2 = -1 \tag{B1}$$

with $u^\mu$ the fluid velocity and $h^\mu$ the direction of the background field, where $\rho$ is its magnitude. To take the zero-field limit smoothly, it is convenient to define the un-normalized vector $B^\mu \equiv \rho h^\mu$ and work to first order in $B^\mu$. Note that in this limit the symmetry of the background is enhanced from $SO(2)$ to $SO(3)$, as the special direction picked out by $h^\mu$ is lost. In particular, note that the transverse $SO(2)$ invariant projector used in [7]

$$\Delta^{\mu\nu} \equiv g^{\mu\nu} + u^\mu u^\nu - h^\mu h^\nu = g^{\mu\nu} + u^\mu u^\nu - \frac{B^\mu B^\nu}{B^2} \tag{B2}$$

is not analytic in $B$; thus we expect that it actually cannot explicitly appear in the zero-field limit. This enforces some restrictions on the form of the hydro theory. E.g. from [7] we have the following form for the first-order dissipative correction to $J^{\mu\nu}$:

$$J_{(1)}^{\mu\nu} = -4r_\perp h^{[\nu} \Delta^{\mu]\beta} h^\rho \nabla_{[\beta} \left( \frac{h_{\rho]}\mu}{T} \right) T - 2r_\parallel \Delta^{\mu\rho} \Delta^{\nu\sigma} \nabla_{[\rho} \left( \frac{\mu h_{\sigma]}}{T} \right) T \tag{B3}$$

where we have set the background sources to zero and rewritten the last term slightly for later convenience. $r_{\perp,\parallel}$ are resistivities that are parallel and perpendicular to the background field; however as the background field is taken to zero, the enhanced symmetry means that these two should coincide, i.e. $r_\perp = r_\parallel \equiv r$. We then find

$$J_{(1)}^{\mu\nu} = -2r\sigma^{\rho[\mu} \sigma^{\nu]\beta} T \nabla_\rho \left( \frac{\mu h_\beta}{T} \right) \tag{B4}$$

where $\sigma^{\mu\nu}$ is the $SO(3)$ invariant projector:

$$\sigma^{\mu\nu} \equiv g^{\mu\nu} - u^\mu u^\nu \tag{B5}$$

and $\Delta^{\mu\nu}$ as defined in (B2) no longer makes an appearance. Now in the small $\rho$ limit we may rewrite

$$\rho = \Xi\mu \tag{B6}$$

where $\Xi$ is the susceptibility $\frac{\partial\rho}{\partial\mu}$, and assembling together (B1) and (B4) the current takes the form

$$J^{\mu\nu} = 2u^{[\mu}B^{\nu]} - 2r\sigma^{\rho[\mu}\sigma^{\nu]\beta}T\nabla_\rho\left(\frac{B_\beta}{\Xi T}\right), \tag{B7}$$

which is manifestly smooth in $B^\mu$.

We now consider a linear perturbation around a fluid at rest (i.e. $u^\mu = \delta^\mu_t$). We work in Fourier space, and give the perturbation spacetime dependence $e^{-i\omega t + ikz}$. We consider a magnetic field perturbation where only $B_x \neq 0$ and where the temperature is held fixed. We find

$$J^{tx} = B_x \qquad J^{zx} = \frac{ikr}{\Xi}B_x \tag{B8}$$

Current conservation $\partial_\mu J^{\mu\nu} = 0$ immediately gives us the dispersion relation

$$\omega = -iDk^2 \qquad D \equiv \frac{r}{\Xi} \tag{B9}$$

which is precisely the mode found holographically in (2.38), modulo the fact that in this section we refer to the resistivity as $r$ (to avoid confusion with the magnetic field density $\rho$) whereas in the main text we refer to the resistivity as $\rho$. Note that dispersion relation cannot be found from taking a direct zero-field limit of the dispersion relations presented in [7], as the hydrodynamic limit taken in that work assumes that the background field is nonzero.

## Appendix C: Wilson and t'Hooft lines in $U(N)$ gauge theory

For completeness, here we review the spectrum of Wilson and t'Hooft lines in $U(N)$ gauge theory. This question is well-studied; recent works include [39, 40]. We found [41] (which studied a similar problem in the context of the Standard Model) particularly helpful and our discussion will follow the approach taken there. Recall first:

$$U(N) = \frac{U(1) \times SU(N)}{\mathbb{Z}_N} \tag{C1}$$

Wilson lines are labeled by $(q, z_e)$, where $q$ is their electric charge under the $U(1)$ and $z_e = 0, 1, \cdots N-1 \in \mathbb{Z}_N$ is their center-valued non-Abelian electric charge. The $\mathbb{Z}_N$ quotient

in the definition of $U(N)$ tells us that allowed Wilson lines have $q = z_e + Nk$, with $k \in \mathbb{Z}$. t'Hooft lines are labeled by $(g, z_m)$, where $g$ is their magnetic charge under the $U(1)$ and $z_m \in \mathbb{Z}_N$.

Mutual locality requires that the Dirac quantization condition between $(q, z_e)$ and $(g', z'_m)$ be satisfied:

$$qg' - \frac{2\pi}{N} z_e z'_m = 2\pi\mathbb{Z} \tag{C2}$$

If we consider $z_e = q = 1$, we find that

$$g' = \frac{2\pi}{N} z'_m + 2\pi p \quad \text{with} \quad p \in \mathbb{Z} \tag{C3}$$

Now we can consider the more general case and check that there are no further restrictions:

$$qg' + \frac{2\pi}{N} z_e z'_m = 2\pi \left( pz_e + z'_m k + pNk \right) = 2\pi\mathbb{Z} \tag{C4}$$

In other words, from the point of view of the $U(1)$ factor alone, minimally quantized t'Hooft lines appear as magnetic-monopoles that carry $1/N$-th the charge of the Dirac monopole. This does not indicate any non-locality; in all observables the phase from the $U(1)$ part cancels against the phase from the non-Abelian part.

For $SU(N)$ gauge theory, the first Abelian term in (C2) is missing. The presence of a minimally charged $\mathbb{Z}_N$ Wilson line $z_e = 1$ sets $z'_m = 0 \pmod{N}$, and thus we have only Wilson lines with no t'Hooft lines.

Similarly, for $SU(N)/\mathbb{Z}_N$ gauge theory, the quotient sets $z_e = 0 \pmod{N}$, and thus the value of $z'_m$ is unconstrained, and we can have any $\mathbb{Z}_N$ t'Hooft line but no Wilson lines.

---

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
