# Peer review of "Generalized global symmetries and holography"

_SciPost Physics, doi:SciPost Phys. 4, 005 (2018)_

## Round 2 · Referee Report · Anonymous · 2017-12-2

Strengths
1) It is a nice paper discussing generalized (higher form) symmetries in holographic models. This subject might seem arcane, but I think it is a formal idea worth fleshing out since there's at least one obvious application to MHD.
2) The paper contains many helpful discussions and "warm-up" calculations in detail. Even though parts of the paper are very technical and I still noted a few parts which I found unclear I thought it was very helpful to include these parts, or sections like 3B would have been very hard to follow.
Weaknesses
1) The only (very minor) weakness is that the paper sometimes feels a bit disjointed -- Sections 2, 3, and 4 all do rather different calculations, although I understand from the ingredients in the action why Sections 2,3,4 are separate and proceed in that order.
Report
This is a largely well-written paper addressing generalized (higher form) symmetries in holographic models. I enjoyed reading this paper since it was a rare example of a current holography paper doing something rather different from others. While I don't know how useful this formalism will end up being, it is still interesting and it should definitely be published.
I would encourage the authors to address the points described below to improve the clarity of the paper, before publication.
Requested changes
1) In (1.9), can I think of the terms b_E F in the action as analogous to j A in a more conventional theory of electromagnetism, but written in a cleaner way? (i.e. *d * b_E ~ j) It seems this is indeed performed in a slightly different setting around (1.20), but I would mention the analogy sooner.
2) Also isn't there a Hodge star required in front of H in either (1.19) or (1.20) since j is a 1-form as defined below (1.17)?
3) Does the simple form of (2.41) follows from a generalized of membrane paradigm for resistivity? I would not be surprised and it should be easy to check. This would also seem to naturally explain the independence from the Landau pole.
4) The pole motion in Figure 1 reminds me of the story of probe brane "zero sound" in holographic models. I'm skeptical, however, that this should be interpreted as a photon, at least in the conventional sense. I'm wondering whether what's happening here is sort of analogous to how conformal sound waves in 1+1D would have c_s = 1, but c_s < 1 in higher d. Maybe this theory in higher d has a slower propagating mode. Probably a better analogy is actually to the holomorphic/anti-holomorphic decoupling in a 1+1D theory which allows for collective charge degrees of freedom to propagate at the speed of light when they would otherwise not do so.
I think resolving this question would be a separate paper as it requires thinking about a higher d bulk theory, but I might change the end of Section 2 to acknowledge this possibility. Maybe in higher d this "photon" travels slower than the speed of light because of an emergent Snell's law in the strongly coupled plasma, for example? Also, the terminology "hydrodynamic-to-collisionless" is maybe not the best -- I know it's conventionally used, but it's more like "hydrodynamic-to-CFT"...
5) I find the discussion in Section 3 to be unclear at points, but some of the topics are on things that I am unfamiliar with so I will give the authors the benefit of the doubt generally. However, I find Figure 5 a bit confusing in its placement. I think it would be easier to first disucss Figure 6, and only introduce Figure 5 afterwards in the discussion on explicitly broken symmetry. Next, I was not clear on whether there is any explicit form (in this model or any others) of the monopole on which \Psi^k can terminate from the point of view of the bulk theory. For example is the explicit form of \phi(x) and A(x) ever known in one of these monopole geometries? The authors should provide a few steps deriving the statement that the B monopole is an insertion of \phi.
6) I see how in 5d bulk holography the B and C are dual to one 2-form current J, but is that true for higher d? The electric-magnetic boundary duality is special to 4d boundary.
7) It looks to me like (4.9) follows from noting that (4.8) can be written as d[.....] = 0, and then using the fact that the resulting closed form changes by an exact form when zeta -> zeta + d\Lambda. I think the authors could re-write (4.8) and (4.9) to make this clearer by writing (4.8) as a total derivative equation, as I did, and then not moving around the i and \lambda in (4.9).

---

## Round 3 · Author Response

We thank the referee for the kind words and for the careful reading of the draft and subsequent detailed suggestions. Below we respond to the specific suggestions made by the referee:

  1. Indeed, in (1.9) b_e can be thought of as an external fixed electric charge density that sources the dynamical Maxwell fields. We agree that it makes sense to discuss this sooner, and have put an extra note below (1.11) emphasizing this when discussing the Maxwell equations that follow from the action.

  2. This is a typo, we thank the referee for catching this and have put back the \star in eq (1.20).

  3. Indeed, the simple form of (2.41) follows from a membrane paradigm-type result; we have added a sentence below (2.39) emphasizing this. We are reluctant however to say that this necessarily provides a “natural” explanation of the independence from the Landau pole, as it does not seem inconceivable that in a more complicated example quantities evaluated at the horizon could also depend on the boundary conditions at infinity through the background Einstein equations.

  4. This is a very good point; we understand the referee’s concerns and have weakened the appropriate paragraphs on page 19 to explain that this is largely an analogy and not a precise statement about universal dynamics. Note that we have kept the “collisionless” terminology though we have modified the parenthetical comment that follows to read "(linearly dispersing)"; while we again completely agree with the referee that the system is not really collisionless, the terminology has now become standard and we feel the referee’s suggestion of “CFT” is also not really ideal in this specific case (as the system is not a CFT in the UV due to the logarithmic running). For most examples of applied holography this would have been a good notation.

  5. Regarding point #5: a. We agree with the referee that it makes more sense to place Figure 6 first; we have thus interchanged the order of Figures 5 and 6 and think that this aids the flow. b. We now address the following question of the referee: "Next, I was not clear on whether there is any explicit form (in this model or any others) of the monopole on which \Psi^k can terminate from the point of view of the bulk theory. For example is the explicit form of \phi(x) and A(x) ever known in one of these monopole geometries? The authors should provide a few steps deriving the statement that the B monopole is an insertion of \phi”. I. As the bulk theory is gapped, the only universal information that exists from the “explicit form” of \phi(x) and A(x) is really the charges of the monopole (all other data characterizing the solution falls off exponentially in space with a correlation length that has been taken to be zero in the limit in which the CS theory applies).
    II. Nevertheless, given the fact that the CS theory is the IR limit of a UV complete theory (that in (3.1)) one could ask if there is any more information present in (3.1) about the monopole. In this case, the statement that \phi(x) is the monopole field then provides a complete specification of the UV core of the monopole, as \phi(x) is a fundamental field that has no internal substructure. This can be contrasted with the situation in the next section, where the “monopole” is a wrapped D5 or NS5 brane, that is not created in a simple way by an elementary field operator in the bulk and it thus possesses quite a lot of internal structure (which is nevertheless in principle fixed by the known stringy physics of these branes). III. As these issues are mostly irrelevant for the mundane purposes of the narrative (which uses only the fact that the monopole “exists”), we chose to not go into these issues in the paper. We have however added a line in the paragraph below Figure 6 on p27 explaining that \phi \sim e^{i\theta} is the monopole field to help explain this connection.

  6. Indeed a similar structure exists in all dimensions; an example is given in the earlier section, where the 4d fields B and A are together dual to only a single 1-form current j_e, but there is no electric-magnetic duality present (or necessary) on the 3d boundary.

  7. We considered this revision: as this is largely a stylistic point on the order of presentation of equations, ultimately we felt that the original derivation was clearer and have chosen to not change the presentation.

You are currently on this page

Resubmission 1707.08577v3 on 16 January 2018

---

## Editorial Decision

published